# Antioxidant and antidiabetic effects of *Angylocalyx oligophyllus* leaves aqueous extract in pregnant diabetic rats: Feto-maternal repercussions

**Christian Tenezogang Takoukam**[1¶], **Marie Claire Tchamadeu**[1*¶],
**Sylvin Benjamin Ateba**[1‡], **William Yousseu Nana**[1‡], **Quelie Selakong Nzekuie**[1‡],
**Armel-Kevin Pechi Fotso**[1‡], **Ahmadou Hassimatou**[1‡], **Calvin Bogning Zangue**[1‡],
**Pascal Emmanuel Owona**[2‡], **Modeste Wankeu-Nya**[1‡], **Alain Bertrand Dongmo**[1‡],
**Dieudonné Massoma Lembè**[1¶]

**1** Department of Biology and Physiology of Animal organisms, Faculty of Science, University of Douala, Douala, Cameroon, **2** Department of Animal Biology and Physiology, Faculty of Science, University of Yaoundé 1, Yaoundé, Cameroon

¶ These authors contributed equally to this work and are Joint Senior Authors
‡ SBA, WYN, CBZ and PEO also contributed equally to this work. QSN, A-KPF and AH also contributed equally to this work. MW-N and ABN also contributed equally to this work.
* marieclaire_tchamadeu@yahoo.fr

## Abstract

Pregestational diabetes mellitus can lead to many adverse outcomes during pregnancy both in the mother and her embryo/fetus. Plant-based products are empirically used as an alternative strategy to reduce these disorders. To investigate the effects of the *Angylocalyx oligophyllus* leaves aqueous extract on diabetes-induced metabolic, reproductive and fetal developmental disorders in pregnant diabetic rats, the in vitro anti-α-amylase and antioxidant plant effects first were evaluated. Then, adult virgin female rats primarily made diabetic by streptozotocin (35 mg/kg) and normal ones were mated with adult male rats. The pregnant rats were distributed into normal and diabetic control groups receiving distilled water, and diabetic rats groups treated with the plant extract doses (50, 100 and 200 mg/kg) or Glibenclamide (standard; 10 mg/kg). Animals were orally treated from 1st to 19th day of gestation, daily weighted, blood glucose levels measured on 1st, 5th, 10th, 15th and 20th gestation days (gd). At the end of pregnancy, maternal diabetic and reproductive parameters, and fetal morphological parameters were analyzed. At the gd 20, there were significant hyperglycemia, altered glucose tolerance, increased total cholesterol, triglycerides, transaminases, liver MDA, SOD, CAT and GSH, reabsorptions sites, post-implantation losses and death fetuses, reduced 17-β-estradiol and numbers of pancreatic cells, corpora luteum, implantation sites and live fetuses in non-treated diabetic mothers, associated with reduced weight and placental and caudal malformations in offsprings. The *A. oligophyllus* leaves aqueous extract induced significant

**Data availability statement:** All relevant data are within the manuscript and its Supporting Information files.

**Funding:** The author(s) received no specific funding for this work.

**Competing interests:** The authors have declared that no competing interests exist.

anti-α-amylase and antioxidant activities *in vitro*. In pregnant diabetic rats, the plant significantly (p < 0.5-p < 0.001) reduced the serum levels of glucose, total cholesterol, triglycerides, LDL-cholesterol, transaminases liver MDA, SOD, CAT and GSH, and post-implantation losses, increased the serum HDL-cholesterol and 17-β-estradiol, the number of pancreatic cells, implantation sites and live fetuses, while reducing placental and caudal malformations, and normalizing fetal weights in offsprings. The *A. oligophyllus* supplementation during pregnancy would be beneficial in preventing reproductive complications related to diabetes mellitus.

## Introduction

Diabetes mellitus is one of the fastest growing public health concerns worldwide affecting nearly 10.5% of the population [1]. It accounts for 35.6% of deaths from non-communicable diseases and 2.7% of deaths from all causes worldwide [2]. Due to an ever-increasing prevalence, more and more women are becoming pregnant with diabetes or developing it during pregnancy, representing approximately 14% of pregnancies worldwide [3].

Both type 1 and type 2 diabetes mellitus (T1DM and T2DM) are significantly associated with female reproductive dysfunction, including amenorrhea or ovulatory abnormalities, hormonal imbalance and decreased ovarian reserve [4,5]. In T1DM, insulin deficiency through the KNDy (kisspeptin, neurokinin B, dynorphin) neural network induces a gonadotropin deficiency (central hypogonadism) and anovulation (infertility), while hyperglycemia accelerates ovarian apoptosis, leading to a reduction in quantity and quality of gametes [6–7].

Diabetes mellitus (T1DM and T2DM) greatly elevates the risk of severe maternal and fetal morbidity and mortality such as spontaneous abortion, fetal demise, preterm delivery, preeclampsia, congenital malformations, hemorrhage, birth trauma, distress syndrome, and even maternal death [8–13]. Events largely related to the degree (timing and quantum of exposure) of hyperglycemia [14] and common and more severe in women with pregestational diabetes (PGDM) as hyperglycemia are more severe and already present before conception [10]. The period around conception and early gestation is a critical and vulnerable window to adverse environmental influences such as hyperglycemia [15]. Throughout this period, hyperglycemia can alter conception and implantation, increase the rate of embryonic resorption and malformation, placental dysfunction, fetal, neonatal and obstetric complications as well as the risk in infants of developing lifetime disease in adulthood [11][16]. Early in pregnancy, hyperglycemia mainly affects the structure of the placenta, while its function is more likely to be affected by the latter disturbances of glucose metabolism [17].

The pathophysiological mechanisms involved in the occurrence and maintenance of all these disorders are not completely understood. Nevertheless, maternal hyperglycemia-related pathogenetic mechanisms including chronic oxidative stress and inflammation have been reported [18–21]. By disrupting electron chain transport in mitochondria, and activating protein kinase C, hexosamine, polyol and advanced

glycation end-product (AGEs) formation pathways, maternal hyperglycemia induces cellular stress through the overproduction of superoxide anion radicals ($O_2^-$) [22,23]. The production of reactive oxygen species decreases the antioxidant defense, increases tissue oxidative stress and also T1DM's complications [24]. In the pancreas, glucotoxicity-induced oxidative stress reduces pancreatic β cell number [25,26], leading to impaired pancreatic function, hyperglycemia and inadequate reproductive outcomes [27]. In the placenta, oxidative stress can impair trophoblast function and disrupt the remodeling of maternal spiral arteries, necessary for placental perfusion and nutrient exchange [28]. By releasing anti-inflammatory factors, uterine CD4[+] regulatory T (Treg) cells support adaptation of uterine vasculature to facilitate placental development [29]. Maternal hyperglycemia-induced oxidative stress reduces uterine number and/or functional competence of Treg cells, altering the maternal immunologic tolerance against the semi-allogeneic fetus and placenta [29,30]. Through a vicious circle, oxidative stress and inflammation perpetually reinforce each other, exacerbating cellular damages and metabolic dysfunction [20,21]. Globally, by impairing early pregnancy implantation and placentation, maternal hyperglycemia-induced oxidative stress and inflammation place women at risk for conditions such as infertility, preeclampsia, recurrent miscarriages, preterm birth, fetal growth restriction [29].

As indicated above, chronic oxidative stress along with inflammation plays a pivotal role in the pathogenesis of pregestational (PGDM) and gestational diabetes mellitus (GDM). Due to their ability to neutralize reactive oxygen species, modulate inflammatory pathways and thus protect cells against oxidative damage, antioxidants hold promise for treating/preventing of gestational diabetes mellitus [31,32]. Therefore, intake of exogenous natural antioxidants (maternal antioxidant supplementation) may support the antioxidant defense [33]. Despite the lack of science-based evidence, the use of herbal products for the management of pregnancy-associated challenges is common [34] in several cultures and areas such sub-Saharan Africa. *Angylocalyx oligophyllus,* is a medicinal plant distributed in tropical Africa (Sierra Leone, Nigeria, Cameroon, Benin, Gabon, Congo, Democratic Republic of Congo and Angola) [35,36] where it is used against various illnesses. In Cameroon, people of Song-Bong (Center Region) traditionally use this plant to treat eye infections and diabetes [37]. Ethnopharmacological information from sellers of natural medicinal products report that the leaves of *A. olligophylus* are used to manage pregnancy even in diabetic women. The phytochemical investigations of this plant led to the identification of several compounds among which formononetin, ursolic acid, betulinic acid, lupeol, lupenone and β-sitosterol [38,39]. The antidiabetic, antihypertension, antioxidant and anti-inflammatory activities of formononetin have been demonstrated in various studies (reviewed by [40]). Zhao et al. [41] furnished compelling evidence for anti-inflammatory and antioxidant properties of ursolic acid. This compound also protected fetal development in pregnant rats with streptozotocin (STZ)-induced GDM [42]. In STZ-induced hyperglycemic mice, lupeol ameliorated the inflammation and apoptosis mediated by the oxidative stress in pancreatic islets [43]. In rats with T2DM, lupenone decreased fasting blood glucose and HbA1c levels, increased hepatic glycogen, and improved oxidative stress and pancreas pathological changes (reduction of islets' number) [44]. It also attenuated endoplasmic reticulum stress and apoptosis in pancreatic beta cells [45]. By decreasing hyperglycemia-mediated glucose intolerance, oxidative stress and inflammation, betulinic acid accelerates diabetic wound healing [46]. In cell model of diabetic nephropathy, β-sitosterol mitigated inflammation, oxidative stress and apoptosis [47]. Based on all this information, *A. oligophyllus* may be beneficial in preventing or treating maternal-fetal complications arising from diabetes mellitus during gestation. Therefore, given that streptozotocin (STZ) is the most commonly used diabetogenic chemical for creating rat models of type 1 and type 2 diabetes and Wistar rats are highly sensitive to it [48], the present study was designed to evaluate the *in vitro* antioxidant and anti-α-amylase activities and the preventive effect of an aqueous extract of leaves of *A. oligophyllus* on maternal-fetal complications in STZ-induced diabetic rats.

## Materials and methods

*In vitro* experiments (anti-α-amylase and antioxidant) were first carried out. The promising results obtained from these assays encouraged *in vivo* experiments. All the study (including the *in-vitro* and *in-vivo* experiments) was conducted over 2-months.

## Ethical approval

In 2021, the Institutional Ethics Committee of the University of Douala accepted this work with the permission number 2512 CEI-UDo/04/2021/T. During the experiments, animals were handled according the directive 2010/63/EU of the European parliament and of the council of 22 September 2010 on the protection of animals used for scientific purposes.

## Drugs and reagents

Iron-chlorid, sodium chloride, Kallium peroxodisulfat, potassium hydroxide were purchased from Acros organics (Germany); Natrium carbonat ($Na_2CO_3$), Tris-HCl and reagent from Roche (Germany); 2-deoxy-D-ribose, sodium hydroxide (NaOH) and sodium nitrite ($NaNO_2$) were obtained from Alfa Aesar (Germany). Phosphate buffered saline (PBS) and NED were purchased from VWR life science (Belgium).

Hydrogen peroxide HR rapid was from Tintometer group GmbH. Trichloroacetic acid and gallic acid were from Cayman chemical company. Alanine aminotransferase (ALT), Aspartate aminotransferase (AST), Total cholesterol, Triglycerides, HDL and total protein (TP) reagent kits were purchased at SGM Italia (Roma).

Trolox (6-hydroxy-2,5,7, 8-tetramethylchromane-2-carboxylic acid), sulfanilamide, Potassium ferricyanide [$K_3Fe(CN)_6$], potassium persulphate, 2,2- Diphenyl-1-picrylhydrazyl (DPPH), thiobarbituric acid, 2,2'-azino-bis-(3-ethylbenzothiazoline-6-sulphonic acid) (ABTS), disodium hydrogen phosphate ($Na_2HPO_4$), phosphoric acid, ferric chloride and ascorbic acid were obtained from Sigma Aldrich (Germany).

## Plant material

The leaves of *Angylocalyx oligophyllus* were harvested on February 12, 2021, from Song-Bong Village (Nyong and Kelle division, Center Region of Cameroon). The species was identified by NANA Victor, a botanical expert at the National Herbarium of Cameroon by comparison with the voucher registered under the number N° 55817/HNC. The leaves were air-dried and then powdered using a grinder.

## Extraction and standard preparation

Preparation and dosage determination of the *A. oligophyllus* leaves aqueous extract were done as previously described [49]. Briefly, 366.6 g of leaves powder were boiled in 2.5 L of distilled water for 20 min. After cooling, the mixture was filtered using a Whatman grade 3 qualitative filter paper. The leaf residue was again decocted following the same procedure. Both filtrates were mixed and evaporated in the oven at 40°C. After evaporation, a mass of 11.813 g of extract was obtained representing a 3.22% extract yield. In that study [49], the 20-day oral administration of the extract at the doses of 50, 100 and 200 mg/kg BW induced no adverse effect on pregnancy, reproduction and fetal growth in nondiabetic pregnant rats, therefore the same doses were chosen and used in the present study to evaluate the effects of the extract on diabetes-induced metabolic, reproductive and fetal developmental disorders in pregnant diabetic rats.

## Phytochemical analysis of *A. olligophyllus* leaves aqueous extract

Phytochemical analysis of *A. olligophyllus* leaves aqueous extract was carried out by qualitative methods, and polyphenol and flavonoids contents of the plant leaf tissue were investigated.

**Qualitative phytochemical analysis.** Qualitative phytochemical screening of the *A. oligophyllus* leaves aqueous extract was carried out following standard procedures as previously described to reveal the presence of saponins (Frotting test), alkaloids (Wagner test), polyphenols ($FeCl_3$ and $K_3Fe(CN)_6$ test), flavonoids and flavonols (Wilstater test), tannins and cathechic tannins ($FeCl_3$, test), triterpenoids (Liebermann-Burchard test), thiol proteins (Biuret test), unsatured steroids (Salkowski test) [50], carotenoids [51] and free quinones (Ether/NaOH test) [52]



**Quantitative phytochemical analysis.** *Total polyphenols determination* The total polyphenolic content in aqueous extract of *A. oligophyllus* leaves was determined according to the Folin-Ciocalteu procedure [53]. About 0.2 mL of *A. oligophyllus* aqueous extract (2 mg/mL) was mixed with 1.2 mL of distilled water and 0.2 mL of Folin-Ciocalteu reagent (10%). After 3 minutes, 0.4 mL of sodium carbonate ($Na_2CO_3$, 7.5%) was added to the mixture which was immediately shaked and incubated for 20 minutes in a water bath at 40°C. Absorbance was measured in a spectrophotometer (UV-Genesis, Germany) at 760 nm and the results were expressed as gallic acid equivalents from a gallic acid standard curve (mg GAE/100g$^{-1}$ Aqueous Extract; $r^2 = 0,943$). The analyses were performed in triplicate.

*Total flavonoid content determination* The total flavonoid content in *A. oligophyllus* leaves aqueous extract was determined following the aluminum chloride colorimetric method of Zhishen et al. [54]. Briefly, a complexe solvent methanol–distilled water–acetic acid was previously prepared (140:50:10, v/v). Then, dried aqueous extract (2 mg) of *A. oligophyllus* leaves was homogenized with the solvent (1 mL) and filtered using Wattman N°3 filter paper. The aluminum chloride ($AlCl_3$) reagent solution was also prepared by dissolving 133 mg of aluminum chloride crystals and 400 mg of sodium acetate crystals in 100 mL of solvent. To 0.2 mL of the extract filtrate was added 1 mL of aluminum chloride ($AlCl_3$) reagent solution and the whole was mixed and incubated for one hour at room temperature. The absorbance was measured at 430 nm using UV-Vis spectrophotometer. The analysis was performed in triplicate. The total flavonoid content was estimated from a quercetin standard curve and the results are expressed as mg quercetin equivalents (mg QE/100g$^{-1}$ Aqueous Extract; $r^2 = 0.9891$).

## Assessment of the *in vitro* anti-α-amylase and antioxidant activities of *A. oligophyllus* leaves aqueous extract
### α-amylase activity inhibition test

The α-amylase inhibition assay was determined according to the modified method of Apostolidis and Lee [55]. Different concentrations (50, 25, 12.5 and 6.125 mg/mL) of *A. oligophyllus* leaf aqueous extract were prepared in 500 µL of phosphate buffer solution (0.02 M; pH 6.9; NaCl 0.006 M) and then, 100 µL of α-amylase solution were added to each tube. After incubation at 25°C for 10 min, 500 µL of starch solution (1% w/v) was added to each tube and the mixture incubated once again (25°C for 10 min). The reaction was stopped by adding 1 mL of 3,5-dinitrosalicylic acid (DNSA) reagent solution. The test tubes were incubated in a water bath at 95°C for 5 min and then cooled to room temperature. In addition, 10 mL of distilled water was added to the reaction mixture and the absorbance was measured at 540 nm against the blank. Acarbose, the standard drug, was used at the same concentrations as the extract. The α-amylase inhibitory activity was expressed as a percentage of inhibition and calculated according to the following formula:

$$\text{Inhibition of } \alpha - \text{amylase } (\%) = \frac{\textbf{Abs standard} - \textbf{Abs sample}}{\textbf{Abs standard}} \times \textbf{100}$$

**DPPH radical scavenging assay.** DPPH free radical scavenging activity of *A. oligophyllus* leaf aqueous extract was assessed using the method described by Mensor et al. [56]. The *A. oligophyllus* extract was dissolved in methanol (1000 µg/mL). Then, 50 µL of this solution was mixed with 150 µL of 0.02% methanolic solution of DPPH to give a final extract concentration ranging from 1000 to 1 µg/mL (1000, 700, 300, 100, 10, 3 and 1 µg/mL). After 30 min incubation in the dark at room temperature, the optical density was measured at 517 nm using a UV/Genesis light spectrophotometer (UV-Genesis, Germany). Ascorbic acid (Vitamin C) was used as a positive control. Each assay was performed in triplicate and results were expressed as percentage of inhibition and calculated according to the following formula:

$$\text{Inhibition of DPPH } (\%) = \frac{\textbf{Abs standard} - \textbf{Abs sample}}{\textbf{Abs standard}} \times \textbf{100}$$

**ABTS scavenging assay.** ABTS⁺ free radical scavenging activity of *A. oligophyllus* leaf aqueous extract was assessed using the method described by Re *et al.* [57]. The *A. oligophyllus* extract was dissolved twice in methanol (1000 µg/mL). 25 µL of the diluted extract was mixed with 65 µL of methanolic solution of 2,2'-azino-bis (3-ethylbenzothiazoline-6-sulphonic acid) (ABTS), to obtain a final concentration of the extract ranging from 1000 to 1 µg/mL (1000, 700, 300, 100, 10, 3 and 1 µg/mL). After 30 min incubation in the dark at room temperature, the absorbances were measured at 734 nm using a UV/Genesis light spectrophotometer (UV-Genesis, Germany). Ascorbic acid (Vitamin C) was used as a control. Each assay was performed in triplicate and results were expressed as a percentage of inhibition and calculated according to the following formula:

$$\textbf{Inhibition of ABTS (\%)} = \frac{\textbf{Abs standard} - \textbf{Abs sample}}{\textbf{Abs standard}} \times \textbf{100}$$

**Hydroxyl scavenging assay.** Hydroxyl radical scavenging generated by the Fenton reaction was measured using the modified protocol of Rao and Kunchandy [58]. To 500 µL of different concentrations (1–1000 µg/mL) of the *A. oligophyllus* extract or Trolox (standard) were added 100 µL of 2-deoxy-D-ribose (28 mM), 100 µL EDTA (1.04 mM), 100 µL FeCl₃ (0.2 mM) (v/v, 1:1), 100 µL H₂O₂ (1 mM) and 100 µL ascorbic acid (1 mM). The mixture (M1) was incubated at 37°C for 1 hour. Thereafter 500 µL of thiobarbituric acid (TBA) (1%) and 500 µL of trichloroacetic acid (TCA) (2.8%) were added to the mixture M1and incubated at 100˚C for 30 min. After cooling and reaching the room temperature, the absorbance of the solution was measured at 532 nm against the blank. Results were expressed as percentage degradation of 2-deoxyribose:

$$\textbf{Inhibition of 2} - \textbf{deoxyribose degradation (\%)} = \frac{\textbf{Astandard} - \textbf{Asample}}{\textbf{Astandard}} \times \textbf{100}$$

**FRAP assay.** The reducing capacity of the extract was performed according to previously described method with modifications [59]. 0.5 mL of extract/ascorbic acid (1–1000 µg/mL) were pipetted and introduced in the test tubes in triplicate long with 1.25 mL of phosphate buffer (200 mM, PH = 6.6) and 1.25 mL of potassium ferricyanide (1%). The mixture was incubated at 50°C for 20 min before adding 1.25 mL of trichloroacetic acid (10%). Following the centrifugation (3000 rpm for 10 min), 1.25 mL of the supernatant was taken away and mixed with 1.25 mL of distilled water and 0.25 mL of FeCl₃ (0.1%). This mixture was incubated for 10 min at 37°C and the absorbance measured at 700 nm against the blank.

## Assessment of the effects of *A. oligophyllus* leaves aqueous extract in pregnant diabetic rats

**Animal material.** The in-vivo study was conducted over 42 days, using 89 Wistar rats (77 female and 12 male) aged 10–12 weeks and weighing 160 ± 20 g. Vaginal smears of adult normal female rats were examined every morning using a pipet tube and 0.9% NaCl, and those exhibiting at least 3 successive and regular estrous cycles were selected for the study. All animals were raised in the animal facility of the Laboratory of Animal Biology and Physiology, Faculty of Science, University of Douala, and housed at room temperature in plastic cages lined with shavings. Rats were housed three or one animals per cage respectively for non-pregnant or pregnant rats. Each housing cage contained information related to the experimental period, as well as the type, dose, and duration of the treatment. Animals were maintained under a natural light/dark cycle and fed a diet consisting of 54% maize meal, 4% wheat meal, 20% fish meal, 10% maize groundnut meal, 3% bone meal, 7% palm kernel oil, 2% salt, and 0.02% vitamin complex. They received tap water as drink *at libitum*.

**Induction of type 1 diabetes in female rats.** The streptozotocin (STZ) solution concentrated at 40 mg/mL was prepared by dissolving 250 mg of STZ powder in 6.25 mL of 0.9% NaCl solution for diabetes induction (STZ from Sigma Chemical Compagny, St Louis, Meule, USA). Type 1 diabetes was induced in 67 normal female rats exhibiting a regular

estrous cycle by intraperitoneal injection of a STZ solution at the dose of 35 mg/kg body weight. Ten another normal female rats with regular estrous cycle used as normal control received the equal volume of the vehicle (0.9% NaCl) [60]. Using an Accu Chek Active glucometer (Sandhofer Strasse, Mannheim, Germany), diabetes was confirmed 72 hours after STZ injection by a fasting blood glucose level > 220 mg/dL [61]. If necessary, rats that did not exhibit hyperglycemia received one or two additional 15 mg/kg injections of STZ solution, administered after diabetes screening on the third and fifth days following the first STZ injection [62].

**Experimental groups and procedure.** Vaginal smears of diabetic and non-diabetic female rats were again examined every morning, and those at the proestrus phase of the estrous cycle were mated overnight with adult male breeder rats (One male with two female rats). The next morning, the observation of spermatozoa in the vaginal smear of a diabetic or non-diabetic female rat was the evidence of the copulation and a probable fertilization, and thus marked the first day (D1) of gestation. The mating period lasted at most 15 days, approximately 3 estrous cycles, allowing to increase the chance of obtaining a large number of pregnant females. During this period, the vaginal smears were examined every morning until fertilization. The non-fertilized female rats, considered as infertile, were excluded from the study [63]. To ensure at least five effectively pregnant rats in each group for reliable statistical analysis by the experiment's end, fertilized rats presumed as pregnant were randomly assigned into six experimental groups based on their initial fasting blood glucose levels as follows:

- Group 1 or normal control (n = 10): non-diabetic pregnant rats receiving distilled water (10 mL/kg).

- Group 2 or diabetic control (n = 20): pregnant diabetic rats receiving distilled water (10 mL/kg);

- Group 3 or positive control (n = 12): pregnant diabetic rats treated with glibenclamide (10 mg/kg);

- Groups 4, 5 and 6 (n = 10 for each group): pregnant diabetic rats receiving the *A. oligophyllus* leaves aqueous extract at the doses of 50, 100 and 200 mg/kg, respectively.

Animals were thus treated by gavage and monitored every day (in the morning) for 20 days. Those that did not exhibit developed pregnancy 12 days after their probable fertilization were removed. Those displaying a sudden and drastic loss of body mass, greatly reduced mobility, extreme fatigue, or abnormal behaviors during the course of the experiment should be euthanized under ketamine and diazepam (80/20 mg/kg) within 24 hours, in case of any recovery. Only effective pregnant normal and diabetic rats were monitored for pregnancy course.

**NB** All removed female rats (those that did not be fertilized over 15-days of mating, and those that did not exhibit developed pregnancy 12 days after their fertilization) were used in another study.

## Pregnancy course monitoring

Maternal body weight and fasting blood glucose levels were measured every 5 days up to the end of pregnancy. Oral glucose tolerance (OGTT) and insulin sensitivity tests were performed on days 17 and 18 of gestation, respectively, as described by Kiss et al. [64]. At the day 20 of pregnancy, pregnant rats were sacrificed under anesthesia (Ketamine/Diazepam complex (70/30)). The whole blood was collected in dry tubes and centrifuged at 3000 rpm for 15 min. The collected serum was separated in two aliquots and stored at −20°C for the measurement of serum 17-β-estradiol and progesterone levels, as well as other serum biochemical parameters. Uterine, liver, kidney, spleen, heart, pancreas, aorta, abdominal fat, pre-gonadic fat, brain, adrenal gland and ovaries were collected and weighed. The uterine horns were dissected to determine the number and examine the morphology of live and/or stillborn pups, and to determine the number of implantation and resorption sites. Placentas were also removed, weighed and their diameter measured. Placental efficiency was calculated according to Wilson and Ford formula [65]. The maternal pancreas and liver portion, as well as placenta were fixed in 10% formalin buffer for histological sections. Another portion from each maternal liver was used for liver oxidative stress markers analysis.

### Oral glucose tolerance test in pregnant normal and diabetic rats

An Oral Glucose Tolerance Test (OGTT) was performed, as described by Kiss et al. [64]. Briefly, on day 17 of gestation, pregnant normal and diabetic rats were weighted and fasted for 06 h. After fasting, animals were administered a single oral dose of D-glucose solution (3 mg/kg) (groups 2–6 or DC, Gli, AoAE50, AoAE100 and AoAE200 groups) or an equal volume of solvent (group 1or NC) by gavage. Blood glucose levels were measured before the D-glucose administration (0 minute), and at 30, 60, and 120 minutes after. At the end of the test, animals received their daily oral treatments by gavage as up described. The area under each curve of glycaemia (AUC) was calculated following GraphPad prism AUC calculation method.

### Insulin sensitivity Test in pregnant normal and diabetic rats

An insulin sensitivity test was performed on day 18 of gestation as described by Kiss et al. [64]. Briefly, non–fasted pregnant normal and diabetic rats were weighed and received a single dose of insulin solution (2 IU/kg) in *i.p.* Blood glucose levels were measured before the insulin administration (0 minute), and at 30 and 60 minutes after. At the end of the test, a glucose solution (0.5 g/kg) was administered to animals in drinking water for 24 hours, in order to counteract possible cases of severe hypoglycemia following the insulin injection [66], allowing sacrificing all animals at the end of experiment without death. Then, daily treatments were administrated orally by gavage in each group and AUC determined as up described.

### Maternal blood glucose level measurement

Blood glucose determination (at 0, 30, 60 and 120 minutes for OGTT; at 0, 30 and 60 minutes for Insulin sensitivity test; and at 0, 5, 10, 15 and 20 days for sub-acute experiment) was carried out using ACCU-CHEK® active glucometer (Roche Diagnostics, Germany) as previously described [66,67].

### Other maternal serum biochemical parameters analysis

After collected in dry tubes, maternal whole blood was centrifuged at 3000 rpm for 15 min and the obtained serum was aliquoted in two tubes and stored at –20°C for the measurement of serum estradiol and progesterone levels, as well as aspartate aminotransferase (AST) and alanine aminotransferase (ALT) activities. Serum concentrations of total protein, total cholesterol (TC), triglycerides (TG), and high-density lipoprotein cholesterol (HDL-C) were also measured, while low-density lipoprotein cholesterol (LDL-C) levels were calculated using Friedewald's formula: LDL-C = (TC) – (HDL-C) – (TG/5) [68].

### Oxidative stress markers analysis in maternal liver

At the end of experiment (Day 20), the liver was removed from each rat, rinsed in 0.9% NaCl solution, wrung out and weighted. A lobe from each maternal liver was ground in Tris-HCl buffer (50 mM; pH = 7.4), centrifuged at 3000 rpm for 15 minutes, and the obtained homogenate (20%) was stored at −20°C in the freezer for assay of the oxidative stress markers (MDA, SOD, CAT and GSH), following the different usually described protocols [69–71].

### Pregnancy outcomes and fetal development

The romoved gravid uteri were dissected for determining the number of live and dead fetuses, reabsorption (embryonic death), implantation sites and of luteal bodies. The number of undetectable implantation sites was determined as described by Costa-Silva et al. [72]. The rate of pre-implantation loss was calculated as [(number of corpora lutea – Number of implantations) x 100/ Number of implantations] [73]. Collected fetuses from the uterine horns were weighed for body weight classification according to the mean ± 1.7 x standard deviation (SD) of body weight obtained in the control group [74]. Placental efficiency defined in percent was calculated according to Wilson and Ford formula [65] as:

$$\text{Placental efficiency} = \frac{\text{Placental mass (g)}}{\text{Real mass of fetus (g)}} \times 100$$

## Statistical evaluation

GraphPad Prism 8.0.1 was used for all statistical evaluations, with differences considered statistically significant at $p < 0.05$.

***In Vitro* Study** For the *in vitro* study, each test was performed in triplicates, and results were expressed as Mean ± Standard Deviation (SD). The Kruskal-Wallis non-parametric test followed by Dunnett's post hoc test was used to analyze the *in vitro* antioxidant and antidiabetic capacities of the plant extract. The IC50 (concentration required to inhibit 50% radicals) values were computed by linear regression of each concentration tested, with the radical scavenging percentage as the response variable.

***In Vivo* Study** Mean values among experimental groups in the *in vivo* study were compared as follows:

- Two-way ANOVA with Bonferroni's post-test was used for comparing repeated measures data between groups, such as body weight and blood glucose.

- One-way ANOVA with Mann-Whitney was used for comparing experimental groups regarding serum biochemical parameters, as well as the number of implantations, live and dead fetuses, resorptions, and corpora lutea.

- Fisher's exact test was used to analyze proportion data, including gestation percentage and fetal mass for gestational age.

## Results

### Qualitative and quantitative phytochemistry of *A. oligophyllus* aqueous extract

"Table 1" shows that thiols proteins, saponins, alkaloids, polyphenolic, flavonoids and flavonols, tannins and cathechic tannins, free quinones, triterpenoids, carotenoids and unsaturated sterols compounds were identified in the *A. olligophyllus* leaves aqueous extract. Polyphenolic compounds being the group mostly explored in qualitative analysis, quantitative analysis of total polyphenols and the sub group flavonoids were also carried out and the results showed that total flavonoids represent ≈ 68.74% of Total polyphenols "Table 1"

### *In vitro* anti-α-amylase and antioxidant activities of *Angylocalyx oligophyllus* leaves aqueous extract

**Anti-α-amylase activity of *A. oligophyllus* leaves aqueous extract.** The plant's capacity to inhibit the α-amylase activity is shown in "Fig 1". The *A. oligophyllus* leaves extract displayed a concentration-dependent inhibitory effect with an inhibitory potency ($IC_{50} = 27.51$ mg/mL) closed to that of acarbose ($IC_{50} = 22.94$ mg/mL). The plant extract concentration of 50 mg/mL highly and significantly inhibited ($p<0.01$) the α-amylase activity (92.07 ± 2.66%) compared to acarbose (82.27 ± 3.28%).

**Free radical scavenging and antioxidant activities of *A. oligophyllus* aqueous extract.** The "Fig 2A" shows the DPPH (2, 2-diphenyl-1-picrylhydrasyl) radical scavenging activity. The aqueous extract of *A. oligophyllus* showed a good capacity to trap the DPPH free radical with an inhibitory concentration 50 (IC50) value of 61.60 µg/mL, while that of ascorbic acid (vitamin C) used as reference was 4.505 µg/mL. The effective concentration 50 ($EC_{50}$) and relative free radical scavenging capacity (RSP) of *A. oligophyllus* extract were 0.97 and 1.02 µg/mL, respectively, compared to those of vitamin C (0.07 and 13.98 µg/mL).

In the ABTS [2, 2'-azino-bis-(3-ethylbenzathiazoline-6-sulfonic) acid] assay, *A. oligophyllus* leaves aqueous extract, like Trolox, showed a concentration-dependent inhibition on ABTS cations "Fig 2B". The extract strongly inhibited $ABTS^+$ radicals with an $IC_{50}$ of 7.097 µg/mL, close to that of Trolox (4.561 µg/mL).

**Table 1. Qualitative and Quantitative Phytochemical Analyses of *A. oligophyllus* leaves aqueous extract.**

| Phytoconstituents | *A. oligophyllus* leaves aqueous extract | | |
|---|---|---|---|
| | Qualitative analyses | Quantitative analyses | |
| | | Total polyphenols (mg GAE/g) | Total flavonoids (mg QE/g) |
| Thiols proteins | + | 145.0±7.5 | 99.68±11.34 |
| Saponins | + | | (≈ 68.74% of total polyphenols) |
| Alkaloids | + | | |
| Polyphenols | + | | |
| Flavonoids | + | | |
| Flavonols | + | | |
| Tanins | + | | |
| Cathechic tanins | + | | |
| Free quinones | + | | |
| Terpenes | / | | |
| Triterpenoids | + | | |
| Carotenoids | + | | |
| Unsatured sterols | + | | |

+ = present; / = not verified; Values are expressed as mean ± SEM; n = 3; mg GAE/g = milli-gram Galic acid Equivalents per gram of extract; mg QE/g = milli-gram Quercetin Equivalents per gram of extract

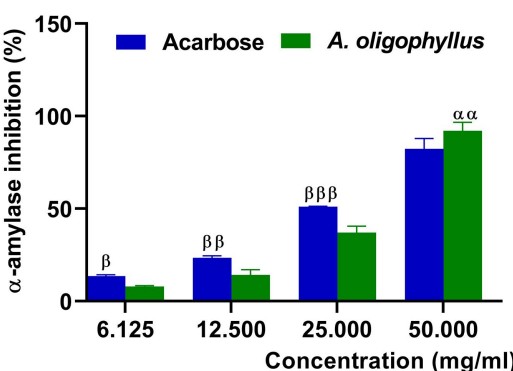

**Fig 1. Effect of different concentrations of *A. oligophyllus* leaves aqueous extract on α-amylase activity.** Each bar represents mean ± SD; n = 3 per group; $^{\alpha\alpha}p < 0.01$ = significant difference to acarbose; $^{\beta}p < 0.05$, $^{\beta\beta}p < 0.01$, $^{\beta\beta\beta}p < 0.001$ = significant difference to plant extract.

In the case of FRAP assay "Fig 2C", the *A. oligophyllus* extract showed good ability to chelate the metal iron with $IC_{50}$ of 36.26 µg/mL, approaching to that of vitamin C ($IC_{50} = 4.61$ µg/mL). However, the chelating power of the plant extract is approximately 8 folds as tower than that of vitamin C ("Fig 2C").

The "Fig 2D" shows the scavenging capacity for hydroxyl radicals. This activity was concentration-dependent for both *A. oligophyllus* aqueous extract and Trolox. The extract $IC_{50}$ value of 25.51 µg/mL was 3.34 times lower than that of the standard antiradical Trolox (85.29 µg/mL). Moreover, the extract showed greater concentration-dependent anti-peroxide activity than Trolox.

### Effects of *A. oligophyllus* leaves aqueous extract in diabetic pregnant rats

**Effects of the plant extract on maternal body and relative organs masses.** The body masses of pregnant normal and diabetic rats significantly increased (p<0.05) during the gestation "Fig 3". The pregnant diabetic rats showed a

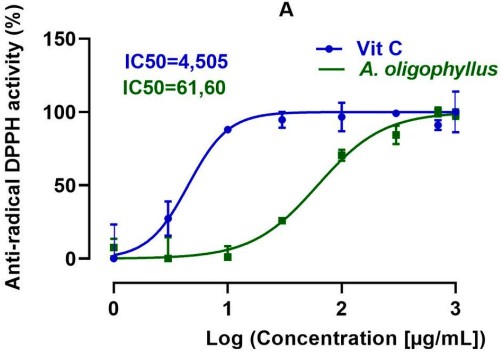

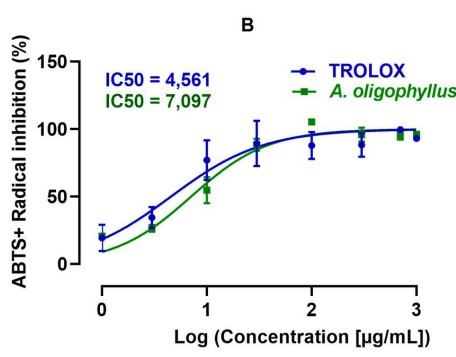

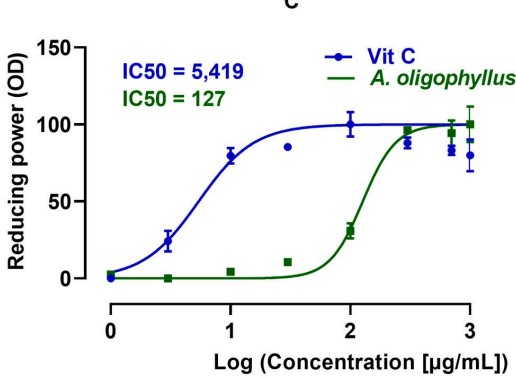

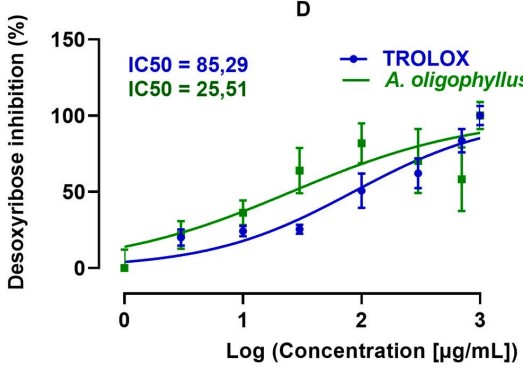

**Fig 2. *A. oligophyllus* extract effects on (A) DPPH; (B) ABTS⁺; (C) FRAP and (D) Hydrogen peroxide.** Each point represents mean ± SD; n = 3 per group.

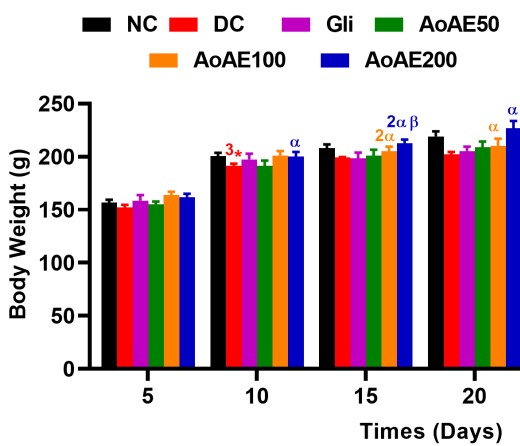

**Fig 3. Body mass changes in pregnant normal and untreated and treated diabetic rats.** Values are expressed as mean ± ESM; n = 5–9 per group; * p < 0.05, ** p < 0.01, ***p < 0.001, = significant difference compared to NC; ᵅp < 0.05, ᵅᵅp < 0.01 = significant difference compared to DC; ᵝp < 0.05 = significant difference from Glib; NC = normal control (non-diabetic pregnant rats); DC = diabetic control (pregnant diabetic rats); Gli = Pregnant diabetic rats treated with glibenclamide; *AoAE50*, *AoAE100* and *AoAE200* = Pregnant diabetic rats treated with *Angylocalyx oligophyllus* aqueous extract at the doses of 50, 100 and 200 mg/kg, respectively.

significant (p<0.001) reduced body mass gain from the 10th day compared to pregnant normal rats. Compared to diabetic control, the *A. oligophyllus* aqueous extract (*Ao*AE) at doses of 100 and 200 mg/kg significantly increased the body mass gain of pregnant diabetic rats from 10th to 20th days (p < 0.05) of gestation, with the maximum increase of 12.18% (p<0.05) observed on day 20 at 200 mg/kg dose. This dose extract (200 mg/kg) also significantly increased the body mass gain of pregnant diabetic rats (7.13%; p<0.05) compared to glibenclamide on day 15 of gestation "Fig 3".

"Table 2" shows in pregnant diabetic control rats significant liver and kidney relative weights increases and abdominal and peri-gonadic fat decreases (p<0.05) as compared to normal control. The administration of the plant extract to pregnant diabetic rats did not significantly decrease the liver relative weight increase, but, significantly decreased the kidney relative weight at the dose extract of 100 mg/kg, as compared to pregnant diabetic control rats (p<0.05) and to glibenclamide-treated ones (p<0.01). In terms of adipose tissue decrease, the *A. oligophyllus* extract at all doses increased the abdominal and peri-gonadic fat in pregnant diabetic rats, with the maximum and significant effects at the dose of 50 mg/kg (by 3.61 and 4.36 folds respectively; p<0.05), compared to pregnant diabetic control. Glibenclamide did not reduce liver and kidney relative masses increase, nor enhance fat masses compared to pregnant diabetic control rats. Moreover, all other organs relative masses did not significantly change between pregnant diabetic control and pregnant normal control rats, despite some increases in pancreas and spleen relative weights observed in glibenclamide- (p < 0.05) and the 200 mg/kg extract dose- (p < 0.05) treated pregnant diabetic groups as compared to pregnant normal control.

**Effect of the plant extract on maternal blood glucose.** The fasting blood glucose remained significantly elevated (p<0.01) in pregnant diabetic rats throughout pregnancy as compared to pregnant normal controls "Table 3". The administration of *A. oligophyllus* aqueous extract at all tested doses decreased this blood glucose from day 5 of gestation compared to pregnant diabetic control. This decrease was significant on days 15 and 20 of gestation at extract doses of 50 mg/kg (39.07% and 45.17% respectively; p<0.01) and 200 mg/kg (48.97%; p<0.001) compared to pregnant diabetic control. Furthermore, the glibenclamide administered to pregnant diabetic rats resulted in non-significant blood glucose decrease of 25.9% at 20th day of gestation compared to pregnant diabetic control.

**Effects of the plant extract on maternal oral glucose tolerance and insulin sensitivity.** Thirty minutes after the administration of D-glucose, a significant increase of 50.91% (p<0.01) in postprandial glycaemia was observed in diabetic

**Table 2. Organs relative masses in pregnant normal and untreated and treated diabetic rats.**

|  |  | Treatments | | | | | |
|---|---|---|---|---|---|---|---|
|  |  | NC (n=9) | DC (n=6) | Gli (n=5) | *AoAE50* (n=8) | *AoAE100* (n=5) | *AoAE200* (n=7) |
| Organ relative mass (g/100 g BW) | Liver | 2.88±0.18 | 3.65±0.20 ** | 3.99±0.22 * | 3.37±0.11 β | 3.26±0.32 | 3.85±0.23 * |
|  | Kidney | 0.46±0.26 | 0.64±0.04 ** | 0.66±0.03 ** | 0.53±0.03 | 0.48±0.04 αββ | 0.68±0.05 ** |
|  | Pancreas | 0.13±0.01 | 0.15±0.02 | 0.18±0.01 ** | 0.15±0.01 | 0.15±0.02 | 0.20±0.03 *** |
|  | Heart | 0.26±0.01 | 0.28±0.01 | 0.27±0.01 | 0.28±0.01 | 0.25±0.02 | 0.31±0.02 * |
|  | Aorta | 0.035±0.002 | 0.041±0.012 | 0.042±0.007 | 0.037±0.007 | 0.033±0.007 | 0.033±0.007 |
|  | Spleen | 0.16±0.02 | 0.28±0.06 | 0.48±0.15 * | 0.23±0.04 | 0.23±0.03 * | 0.32±0.06** |
|  | Abdominal fat | 1.18±0.21 | 0.23±0.09 ** | 0.37±0.19 * | 0.83±0.26 α | 0.95±0.25 | 0.52±0.16 * |
|  | Peri-ovarian fat | 1.28±0.11 | 0.28±0.11 ** | 0.53±0.21 * | 1.22±0.17 α | 0.60±0.09 ** | 1.10±0.21α |
|  | Brain | 0.71±0.05 | 0.80±0.02 | 0.85±0.04 | 0.70±0.07 | 0.72±0.06 | 0.82±0.02 |
|  | Adrenal gland | 0.031±0.004 | 0.035±0.003 | 0.042±0.003 | 0.028±0.002 | 0.033±0.004 | 0.036±0.004 |
|  | Ovaries | 0.036 ±0.002 | 0.041±0.003 | 0.042±0.003 | 0.037±0.003 | 0.033±0.003 | 0.033±0.003 |

Values are expressed as mean±SEM; n=5–9; *p<0.05, **p<0.01, ***p<0.001=significant difference compared to NC; αp<0.05=significant difference compared to DC; βp<0.05, ββp<0.01=significant difference compared to Gli; NC=normal control; DC=diabetic control; Gli=Pregnant diabetic rats treated with glibenclamide; *AoAE50*, *AoAE100* and *AoAE200*=Pregnant diabetic rats treated with *Angylocalyx oligophyllus* aqueous extract at the doses of 50, 100 and 200 mg/kg, respectively.

**Table 3. Blood glucose levels variation in pregnant normal and untreated and treated diabetic rats.**

| Treatments | Times (Days) | | | | |
|---|---|---|---|---|---|
| | D1 | D5 | D10 | D15 | D20 |
| NC (n=6) | 107.70±7.68 | 99.20±4.98 | 87.17±4.25 | 91.83±4.79 | 86.67±4.73 |
| DC (n=5) | 350.40±48.16** | 391.00±2.19**** | 360.20±41.30*** | 372.20±37.64**** | 397.20±11.43**** |
| Gli (n=5) | 366.00±44.03 | 359.20±55.26 | 353.20±28.22 | 354.40±49.64 | 294.20±26.16 |
| *AoAE50* (n=6) | 331.00±34.36 | 304.50±93.75 | 317.00±37.21 | 226.80±49.45 ᵅᵅ | 217.80±33.21 ᵅᵅ |
| *AoAE100* (n=5) | 340.20±42.75 | 250.40±48.12 | 292.60±20.30 | 298.80±53.91 | 263.00±52.25 |
| *AoAE200* (n=6) | 317.20±22.38 | 272.70±30.20 | 282.20±48.63 | 253.70±38.95 | 202.70±40.73ᵅᵅᵅ |

Values are expressed as mean±SEM; n=5–6 per group; *p<0.05, **p<0.01, ***p<0.001, ****p<0.001=significant difference compared to NC; ᵅp<0.05, ᵅᵅp<0.01, ᵅᵅᵅp<0.001=significant difference compared to DC; NC=normal control; DC=diabetic control; Gli=Pregnant diabetic rats treated with glibenclamide; *AoAE50*, *AoAE100* and *AoAE200*=Pregnant diabetic rats treated with *Angylocalyx oligophyllus* aqueous extract at the doses of 50, 100 and 200mg/kg, respectively.

control rats compared to normal control "Fig 4A". The administration of the plant extract at the dose of 200mg/kg reduced this blood glucose increase by 21.85% compared to pregnant diabetic control and moreover by approximately 1.5 folds compared to glibenclamide-treated diabetic rats group. The glibenclamide reduced the pregnant diabetic rats' postprandial glycaemia by 15.1% compared to pregnant diabetic control. However, the estimation of the area under the glycaemia curve (AUC) for each group, expressing mean blood glucose levels between 0 and 120min, showed high and significant (p<0.01) blood glucose levels in diabetic groups than in the normal control group after carbohydrate loading. The plant extract doses of 100 and 200mg/kg respectively inhibited the global postprandial glycaemia increase (AUC) by 9.5% and 16.1% (p<0.05) compared to the pregnant diabetic control "Fig 4A".

"Fig 4B" shows insulin sensitivity in diabetic pregnant rats. The intra-peritoneal injection of insulin in pregnant diabetic rats resulted in a blood glucose levels decrease in all groups. However, 60 minutes after insulin injection, the blood

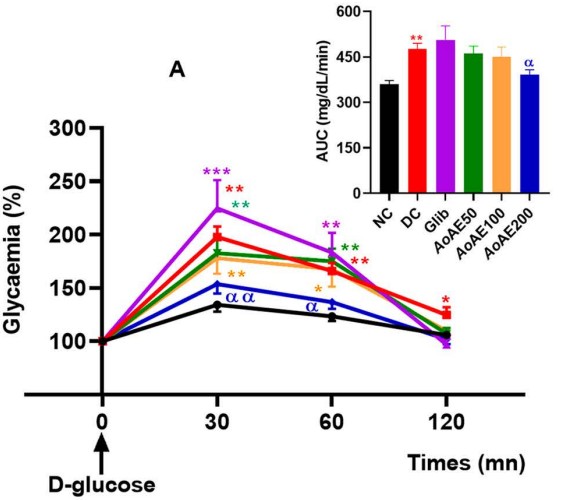
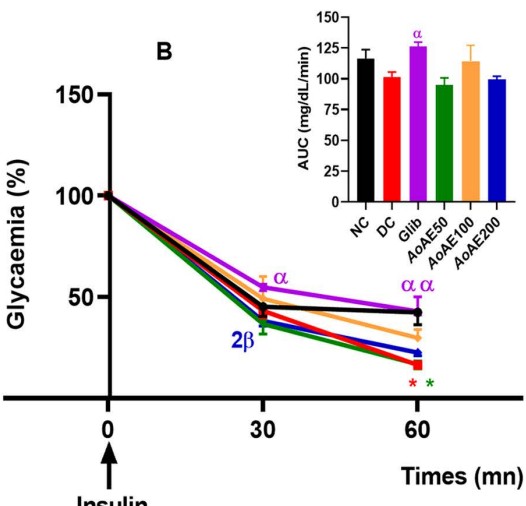

**Fig 4. Oral glucose tolerance (A) and insulin sensitivity (B) in *A. oligophyllus* extract-treated pregnant diabetic rats.** Values are expressed as mean±SEM; n=5–6 per group; ** p<0.01, ****p<0.0001=significant difference Compared to NC; ᵅp<0.05,=significant difference compared to DC; AUC=Area Under the glycaemia Curve; NC=normal control; DC=diabetic control; Gli=Pregnant diabetic rats treated with glibenclamide; *AoAE50*, *AoAE100* and *AoAE200*=Pregnant diabetic rats treated with *Angylocalyx oligophyllus* aqueous extract at the doses of 50, 100 and 200mg/kg, respectively.

glucose levels reduction was greater in pregnant diabetic control rats (83.6%; p < 0.001) than in pregnant normal control rats (42.5%; p < 0.001) as compared to the initial blood glucose values, which resulted in a reduced AUC values in diabetic control (50.73 mg/dL/min) compared with normal control (58.21 mg/dL/min). The plant extract at 100 mg/kg dose and glibenclamide respectively reduced the diabetic pregnant rats' hypersensitivity (AUC) by 12.5% and 24.46% compared to pregnant diabetic control.

**Effects of the plant extract on some of maternal biochemical parameters.** Pregnant diabetic rats showed a significant increase (p<0.05) in serum transaminases (ALT, AST and the AST/ALT ratio), triglycerides, total cholesterol, LDL-cholesterol, and a significant decrease (p<0.05) in HDL-cholesterol compared to normal control rats "Table 4".

The *A. oligophyllus* aqueous extract at all tested doses reduced serum AST and ALT activities as well as AST/ALT ratio, with significant reductions observed at the 100 mg/kg dose for ALT (42.1%%; p<0.05), at doses of 50, 100 and 200 mg/kg for AST (48.7%, 36% and 46.8% respectively; p<0.05), and at doses of 50 and 200 mg/kg for AST/ALT ratio (17.04% and 36.63% respectively; p<0.05), all compared to the pregnant diabetic control "Table 4". In addition, the plant extract at doses of 50, 100 and 200 mg/kg also reduced (p<0.05) serum levels of triglycerides (inversely dose-dependently by 36.1%, 33.1% and 24% respectively), total cholesterol (dose-dependently by 36%, 41.9% and 47.7% respectively) and LDL-cholesterol (Inversely dose-dependently by 42%, 33.7% and 12% respectively), while increasing HDL-cholesterol levels (dose-dependently by 99.5%, 146.9% and 258.8% respectively; p<0.05), as compared to pregnant diabetic control. Interestingly, the plant extract doses of 100 and 200 mg/kg increased the HDL-cholesterol level by 5% and 52.54% respectively, compared with pregnant normal control rats.

"Table 4" shows that the glibenclamide also induced significant reductions in serum AST (38.1%; p<0.05) and ALT (51.6%; p<0.5) activities, and total cholesterol levels (42%; p<0.01), while increasing HDL-cholesterol levels (100.8%; p<0.05) and AST/ALT ratio (33.05%; p>0.05) compared to pregnant diabetic control.

**Table 4. Maternal serum and liver biochemical parameters in pregnant non-diabetic and untreated and treated diabetic rats.**

| Parameters | Treatments | | | | | |
|---|---|---|---|---|---|---|
| | NC (n=5) | DC (n=5) | Gli (n=5) | *AoAE50* (n=5) | *AoAE100* (n=5) | *AoAE200* (n=5) |
| **Hepatic Serum Biomarkers** | | | | | | |
| ALT (U/L) | 30.1±5.4 | 56.8±9.2 * | 27.5±6.9 ᵃ | 33.0±7.7 | 32.9±4.3 ᵃ | 40.7±2.4 |
| AST (U/L) | 106.0±16.9 | 281.4±31.2 *** | 174.2±25.2 ᵃ | 144.4±14.9 ᵃᵃ | 180.0±10.7 ᵃ | 149.7±17.2 ᵃᵃ |
| AST/ALT | 3.66±0.5 | 5.87±1.5 ** | 7.81±2.1 | 4.87±0.6 | 5.75±0.5ᵃ | 3.72±0.5 ᵃᵃ |
| Total Proteins (g/L) | 3.3±0.2 | 3.1±0.2 | 3.6±0.3 | 3.6±0.3 | 3.5±0.2 | 4.4±0.2 ** ᵃᵃ |
| Triglycerides (mg/dL) | 133.1±9.5 | 192.5±9.9 ** | 153.5±16.7 | 123.1±11.0 ᵃᵃ | 128.7±4.4 ᵃ | 146.3±6.8 ᵃ |
| Total Cholesterol (mg/dL) | 65.22±5.28 | 109.0±8.4 * | 63.23±6.71 ᵃᵃ | 69.75±9.27 ᵃ | 63.28±7.78 ᵃᵃ | 60.25±3.05 ᵃᵃᵃ |
| HDL-Cholesterol (mg/dL) | 17.15±3.17 | 7.29±0.11 ** | 14.64±1.20 ᵃ | 14.54±1.14 ᵃᵃ | 18.00±2.65 ᵃ | 26.16±2.34 * ᵃ β |
| LDL-Cholesterol (mg/dL) | 80.49±9.62 | 116.5±7.14 * | 96.78±11.15 | 67.59±10.2 ᵃ | 77.19±14.52 | 102.5±5.76 |
| **Liver Tissue Biomarkers of Oxidative Stress** | | | | | | |
| MDA (µmol/mg of protein) | 0.007±0.001 | 0.018±0.003 ** | 0.009±0.002 ᵃ | 0.008±0.000 ᵃᵃ | 0.006±0.001 ᵃᵃ | 0.007±0.001 ᵃᵃ |
| SOD (U/mg of protein) | 0.423±0.119 | 1.958±0.216 ** | 1.682±0.083 | 0.699±0.080 ᵃᵃ | 0.631±0.196 ᵃᵃ | 0.767±0.092 ᵃᵃ |
| CAT (µmol of $H_2O_2$/min/mg of protein) | 0.327±0.040 | 0.888±0.132 ** | 0.352±0.060 ᵃ | 0.460±0.051 ᵃ | 0.392±0.030 ᵃᵃ | 0.362±0.042 ᵃᵃ |
| GSH (µmol/mg of protein) | 0.076±0.008 | 0.117±0.008 * | 0.096±0.012 | 0.075±0.006 ᵃᵃ | 0.049±0.004 ᵃᵃ | 0.050±0.008 ᵃᵃ |

Values are expressed as mean±SEM; n=5 per group; *p<0.05, **p<0.01, ***p<0.001=significant difference compared to NC; ᵃp<0.05, ᵃᵃp<0.01=significant difference compared to DC; βp<0.05=significant difference from Gli; NC=normal control; DC=diabetic control; Gli=Pregnant diabetic rats treated with glibenclamide; *AoAE50*, *AoAE100* and *AoAE200*=Pregnant diabetic rats treated with *Angylocalyx oligophyllus* aqueous extract at the doses of 50, 100 and 200 mg/kg, respectively; ALT=ALanine Transaminase; AST=Aspartate Transaminase; HDL-Cholest.=High-Density Lipoprotein cholesterol; LDL-Cholest.=Low-Density Lipoprotein cholesterol; MDA=Malondialdehyde; SOD=Superoxide Dismutase; CAT=Catalase; GSH=Reduced Glutathione.

"Table 4" also shows that compared to non-diabetic pregnant rats, the pregnant diabetic control rats exhibited significant increases (p < 0.01) in liver activities/levels of malondialdehyde (MDA), superoxide dismutase (SOD), Catalase (CAT), and reduced Glutathione (GSH). The administration of the aqueous extract of *A. oligophyllus* at doses of 50, 100 and 200 mg/kg significantly decreased (p < 0.01) the elevated levels of MDA (by 56.2%, 68% and 60.7% respectively) and GSH (by 36.3%, 58.3% and 57% respectively) in the liver of treated pregnant diabetic rats, compared to pregnant diabetic control. The plant extract doses of 50,100 and 200 mg/kg also decreased the elevated liver activities of SOD (respectively by 64.3%, 67.8% and 60.8%; p < 0.01) and catalase (inversely dose-dependently by 48.2%, 55.9% and 59.2%; p < 0.05) in treated pregnant diabetic rats, compared to pregnant diabetic control. Glibenclamide only reduced (p < 0.05) the elevated liver MDA (50%) and CAT (60.36%) levels in pregnant diabetic rats, compared to pregnant diabetic control.

**Effect of the plant extract on histomorphological changes in maternal pancreas and liver.** Microscopic examination of pancreatic sections from pregnant normal control rats "Fig 5" showed a regular appearance of islets of Langerhans resembling a rounded or oval pale-colored area not encapsulated within the pancreatic lobules, which are made up of groups of cells arranged in irregular, branched and anastomosing cords separated by blood capillaries. However, the pregnant diabetic rats' pancreatic sections showed atrophied and shrunken islets of Langerhans. On the other hand, *A. oligophyllus* extract and glibenclamide increased islet volume with near-normal contours showing cellular restoration in treated pregnant diabetic rats "Fig 5". The "Fig 6" shows that pre-gestational diabetes significantly reduced (p<0.05) the number of pancreatic cells by 35.49% compared to normal control. The pancreatic cells number of diabetic rats treated with the plant extract doses of 50, 100 and 200 mg/kg and glibenclamide significantly increased (p<0.05), respectively by 58.44%, 57.96%, 42.80% and 54.66%, compared to diabetic control.

"Fig 7" shows the microphotographs of the liver structure. It is evident from this figure that normal pregnant rats presented a normal hepatic parenchyma with a well-differentiated portal vein, hepatic artery and bile duct. On the other hand, untreated pregnant diabetic rats presented leukocyte infiltrations and vascular congestion. Furthermore, pregnant diabetic rats treated with glibenclamide and plant extract at a dose of 200 mg/kg also presented mild inflammation and vascular congestion. The plant extract at doses of 50 and 100 mg/kg improved these alterations in pregnant diabetic rats.

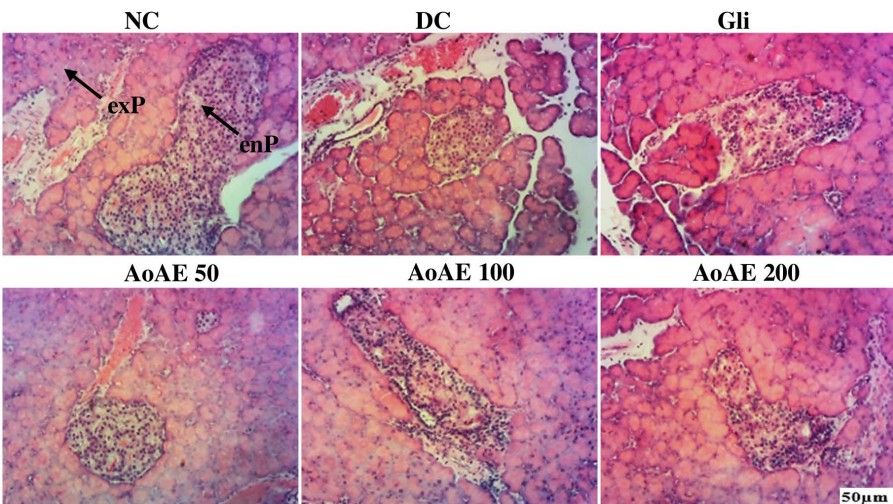

**Fig 5. Microphotography of pancreas sections (haematoxylin-eosin x100) from pregnant non-diabetic (NC) and untreated (DC) and treated diabetic rats.** NC = normal control; DC = diabetic control; Gli = Pregnant diabetic rats treated with glibenclamide; *AoAE50*, *AoAE100* and *AoAE200* = Pregnant diabetic rats treated with *Angylocalyx oligophyllus* aqueous extract at the doses of 50, 100 and 200 mg/kg, respectively; exP = exocrine pancreas; enP = endrocrine pancreas, Red arrow = reduced diameter; Black arrow = normal diameter.

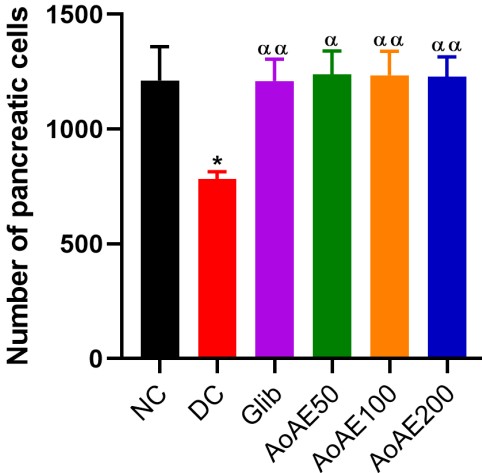

**Fig 6. Histomorphometrical data of pancreas sections from pregnant non diabetic and untreated and treated diabetic rats.** Values are expressed as mean ± SEM; n = 5 per group; *p < 0.05 = significant difference compared to NC; $^{\alpha}$p < 0.05, $^{\alpha\alpha}$p < 0.01 = significant difference compared to DC; NC = normal control; DC = diabetic control; Gli = Pregnant diabetic rats treated with glibenclamide; *AoAE50*, *AoAE100* and *AoAE200* = Pregnant diabetic rats treated with *Angylocalyx oligophyllus* aqueous extract at the doses of 50, 100 and 200 mg/kg, respectively.

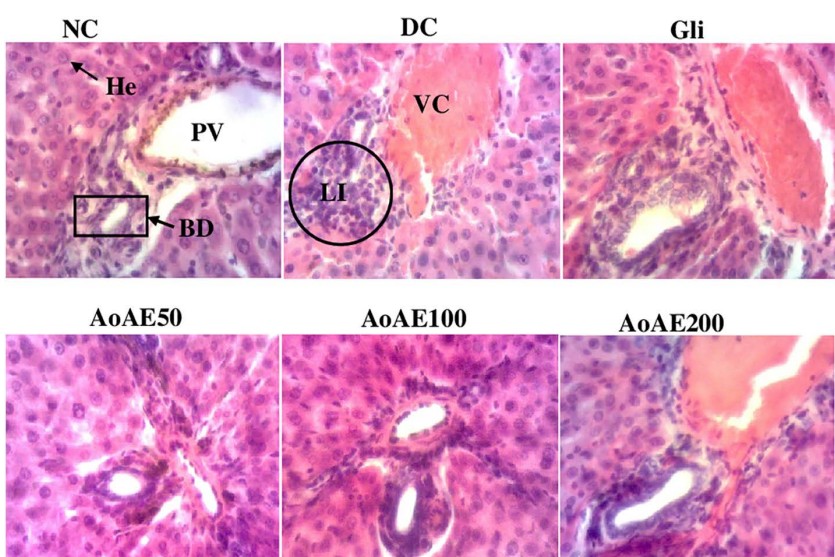

**Fig 7. Microphotography of liver sections (haematoxylin-eosin x100) from pregnant non diabetic and untreated and treated diabetic rats.** NC = normal control; DC = diabetic control; Gli = Pregnant diabetic rats treated with glibenclamide; *AoAE50*, *AoAE100* and *AoAE200* = Pregnant diabetic rats treated with *Angylocalyx oligophyllus* aqueous extract at the doses of 50, 100 and 200 mg/kg, respectively; He = Hepatocyte; PV = Portal Vein, LI = Leukocyte Infiltration; BD = Bile Duct; VC = Vascular Congestion.

**Effects of the plant extract on maternal reproductive parameters.** Maternal reproductive parameters registered at the 20th day of pregnancy are shown in "Table 5". Overall, fertilization and pregnancy outcomes presented interesting findings in this study.

In terms of fertilization, a high fertilization rate of 93.5% was observed among the 77 female rats (10 normal and 67 diabetic) mated. This included all 10 normal females and 62 diabetic females. A small proportion (6.5% or 5 total female rats), all of whom were diabetic, remained unfertilized after 15 days of mating and were categorized as infertile.

**Table 5. Treatment-related variations in reproductive parameters of pregnant rats.**

| Parameters | Groups | | | | | |
|---|---|---|---|---|---|---|
| | NC (n=8) | DC (n=6) | Gli (n=5) | *AoAE50* (n=7) | *AoAE100* (n=5) | *AoAE200* (n=7) |
| Pregnant (Day 0) (N) | 10 | 20 | 12 | 10 | 10 | 10 |
| Effectively pregnant (Day 20) (N) | 9 | 6 | 5 | 8 | 5 | 7 |
| % Pregnancy | 90 | 30 ** | 41.67 * | 80 ᵅ | 50 | 70 ᵅ |
| Gestational index (%) Mean±SEM | 85.0±5.0 | 35.0±6.4 ** | 55.6±6.3 | 80.0±4.5 ᵅ | 76.7±7.1 ᵅ | 80.0±7.6 ᵅ |
| Corpora lutea (N) Mean±SEM | 265 33.13±2.62 | 122 ** 20.33±2.32 | 169 33.80±2.22 | 203 ᵅ 29.0±2.14 | 165 ᵅ 33.0±1,95 | 165 ᵅ 27.0±3.55 |
| Implantation (N) Mean±SEM | 74 9.3±0.9 | 48 8.0±0,5 | 39 7.8±0,5 | 59 8.4±0.4 | 38 7.6±3.1 | 63 9.0±0.6 |
| Pre-implantation loss (%) Mean±SEM | 566.1 70.8±3.5 | 406.2 * 58.0±6.4 | 384.2 ᵅᵅ 76.8±0.9 | 487.9 69.7±3.1 | 382.4 ᵅᵅ 76.5±3.1 | 385.6 64.3±5.8 |
| Post-implantation loss (%) Mean±SEM | 8.3 1.8±1.8 | 131.3 * 18.8±13.7 | 54.17 10.8±2.8 | 12.5 ᵝ 1.79±1.8 | 14.3 2.9±2.9 | 65.4 9.3±4.7 |
| Resorptions (N) Mean±SEM | 1 0.13±0.13 | 9 * 1.28±1.2 | 2 0.33±0.21 | 0 0.00±0.00 | 0 0.00±0.00 | 1 0.14±0.14 |
| Fetal survival index (%) Mean±SEM | 95.2±7.5 | 82.8±13.9 | 91.7±3.5 | 94.8±2.4 | 97.6±2.4 | 89.9±5.3 |
| Live fetuses (N) Mean±SEM | 73 9.1±0.8 | 38 * 6.3±1.3 | 35 7.0±0.6 | 58 8.3±0.4 | 46 7.7±0.7 | 57 8.1±0.7 |
| Dead fetuses (N) Mean±SEM | 0 0.0±0.0 | 8 * 1.1±1.4 | 0 0.0±0.0 | 0 0.0±0.0 | 0 0.0±0.0 | 0 0.0±0.0 |
| Uterine horn (g) | 39.0±2.2 | 27.2±3.5 * | 24.7±1.0 | 39.0±2.9 ᵅᵝᵝ | 29.2±2.6 | 32.0±1.9 ᵝ |

Values are expressed as mean±SEM; n=5–8 per group; *p<0.05, **p<0.01=significant difference compared to NC; ᵅp<0.05, ᵅᵅp<0.01=significant difference compared to DC; ᵝp<0.05, ᵝᵝp<0.01=significant difference from Gli; NC=normal control; DC=diabetic control; Gli=Pregnant diabetic rats treated with glibenclamide; *AoAE50*, *AoAE100* and *AoAE200*=Pregnant diabetic rats treated with *Angylocalyx oligophyllus* aqueous extract at the doses of 50, 100 and 200 mg/kg, respectively.

For pregnancy outcomes (among the fertilized population), 51.9% (40 out of the 77 total, or 9 normal and 31 diabetic) achieved an effective pregnancy. Intriguingly, 41.6% of the fertilized rats (32 total, or 1 normal and 31 diabetic) did not develop a pregnancy 12 days post-fertilization, highlighting a significant difference in pregnancy success, particularly among the diabetic group.

Furthermore, statistical analysis in each experimental group revealed that the coexisting adverse effects of pregnancy and pregestational diabetes in pregnant diabetic control rats, when compared to pregnant normal control rats, led to significant reductions (p<0.05) in gestation percentage (66.67%), pre-implantation loss (28.25%), corpora lutea (53.96%), born fetus numbers (47.95%), and uterine horn mass (30.26%). Additionally, there were significant increases (p<0.05) in post-implantation losses (1482% or ≈ 15.82-fold), resorptions (800% or ≈ 9-fold), and the number of stillborn fetuses (8) in the pregnant diabetic control group compared to the pregnant non-diabetic rat group.

Administration of *A. oligophyllus* leaves aqueous extract to pregnant diabetic rats increased the gestation percentage, significantly (p<0.05) by 2.67 and 2.33-fold (166.67%, and 133.33%) at doses of 50 and 200 mg/kg respectively, and non-significantly by 1.67 (66.67%) at 100 mg/kg dose extract. The plant extract also increased but non-significantly born fetus numbers by 52.63%, 21.05%, and 50% at respective doses of 50, 100 and 200 mg/kg. However, it significantly increased (p<0.05) corpora lutea numbers by 66.39%, 35.25%, and 35.25% at these same respective doses, all compared to pregnant diabetic control rats. In pregnant diabetic rats, only the 100 mg/kg dose of the plant extract significantly reduced the pre-implantation loss (5.86%; p<0.01) compared to pregnant diabetic control rats. Interestingly, the plant extract at all tested doses (50–200 mg/kg) reduced the post-implantation loss respectively by 90.48%, 89.11% and

50.19%, and prevented the resorptions and fetal death, as compared to pregnant diabetic control rats. Furthermore, the plant extract increased the uterine horn mass at all doses but significantly at a dose of 50 mg/kg (43.38%; p<0.05) compared to pregnant diabetic control rats, and at doses of 50 mg/kg (56%; p<0.01) and 200 mg/kg (29.7%; p<0.05) compared to glibenclamide-treated rats' group.

In glibenclamide-treated pregnant diabetic rats, compared to pregnant diabetic control, there were reduced pre-implantation loss (5.42%; p<0.01), post-implantation loss (58.74%), resorptions (77.78%), and live fetus numbers (7.89%). Additionally, no stillborn fetuses were recorded, and there was an increased gestation percentage (11.67%) and corpora lutea number (38.52%).

**Effects of the plant extract on maternal serum 17-β-Estradiol and Progesterone levels.** The "Fig 8A" shows that serum 17-β-estradiol levels significantly decreased by 27.54% (p<0.01) in pregnant diabetic control rats compared to pregnant normal control. The *A. oligophyllus* leaves aqueous extract at all doses tested significantly increased (p<0.01) serum 17-β-estradiol levels with maximum percentages observed at doses of 50 mg/kg (50.38%) and 200 mg/kg (40.14%) compared to pregnant diabetic control.

In the other hand, slight non-significant variations in serum progesterone levels were recorded between groups "Fig 8B", with a 1.8% decrease in pregnant diabetic control compared to pregnant normal control, and enhancements of 8.5%, 3.1% and 5.5% observed at the plant extract doses of 50, 100 and 200 mg/kg compared to pregnant diabetic control. The glibenclamide induced a non-significant 22.94% increase in serum 17-β-estradiol level, but did not vary the progesterone level compared to pregnant diabetic control.

### Effects of the aqueous extract of *A. oligophyllus* leaves on fetal parameters

**Effects of the plant extract on fetal weight and external morphological characteristics.** "Table 6" shows the equivalent fetuses weight percentages of gestational age between experimental groups. The pups from diabetic control rats had a low weight (p<0.05) compared to those from non-diabetiic or normal control rats. In addition, the AGA (adequate fetal weight for gestational age) percentage decreased (p<0.05), whereas percentages of SGA (small fetal weight for gestational age) (p<0.05)

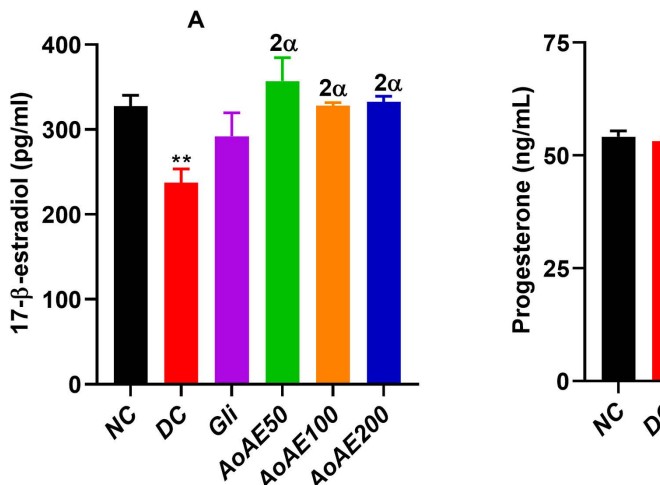
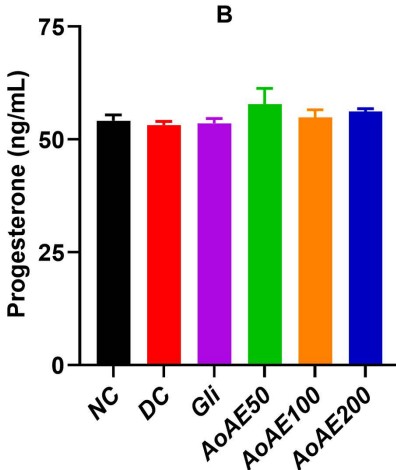

**Fig 8. Serum17-β-estradiol (A) and progesterone (B) levels in pregnant normal and untreated and treated diabetic rats.** Values are expressed as mean±SEM; n=5 per group; *p<0.05, **p<0.01 = significant difference compared to NC; ααp<0.01 = significant difference compared to DC; NC=normal control; DC=diabetic control; Gli=Pregnant diabetic rats treated with glibenclamide; *AoAE50*, *AoAE100* and *AoAE200*=Pregnant diabetic rats treated with *Angylocalyx oligophyllus* aqueous extract at the doses of 50, 100 and 200 mg/kg, respectively.



**Table 6. Variation in fetal mass and percentages of fetal weight corresponding to gestational age, according to treatments.**

| Groups | Parameters | | | |
|---|---|---|---|---|
| | Fetal body weight (g) | SGA (%) | AGA (%) | LGA (%) |
| NC (n=8) | 3.2±0.2 | 0.0 | 92.65 | 7.35 |
| DC (n=6) | 2.1±0.2 * | 40.81 * | 48.98 * | 10.20 |
| Gli (n=5) | 3.2±0.4 ᵃ | 7.5 | 70 | 22.5 * |
| *AoAE50* (n=7) | 3.1±0.2 ᵃᵃ | 0.0 ᵃ | 97.96 ᵃ | 2.04 |
| *AoAE100* (n=5) | 2.9±0.5 | 4.35 | 78.26 | 17.39 |
| *AoAE200* (n=7) | 3.3±0.4 ᵃ | 3.57 | 50 | 46.43 ᵃᵝ |

Values are expressed as mean±SEM; n=5–8 per group; * p<0.05, = significant difference compared to NC; ᵃp<0.05, ᵃᵃp<0.01 = significant difference compared to DC; ᵝp<0.05 = significant difference from Gli; NC=Normal control; DC=diabetic control; Gli=Pregnant diabetic rats treated with glibenclamide; *AoAE50*, *AoAE100* and *AoAE200* = Pregnant diabetic rats treated with *Angylocalyx oligophyllus* aqueous extract at the doses of 50, 100 and 200mg/kg, respectively. SGA=Small Gestational Age; AGA=Adequate Gestational Age; LGA=Large Gestational Age.

and LGA (larger fetal weight for gestational age) increased in diabetic control group compared to normal control. However, the percentages of SGA and AGA from the diabetic control group were almost equal (40.81% vs. 48.98%).

The *A. oligophyllus* extract and glibenclamide normalized the pups body weights (p<0.05-p<0.01). Furthermore, although the SGA percentages were very low in pregnant diabetic rats treated with *A. oligophyllus*, the distribution of pups in these groups between AGA and LGA was such that the AGA increased and LGA decreased in an inversely dose-dependently manner, with significant values (p<0.05) at the 50mg/kg dose, AGA and LGA percentages being almost equal in the 200mg/kg dose (50% vs. 46.43%), all compared to the diabetic controls. Moreover, the plant extract doses of 100 and 200mg/kg showed high percentages of LGA compared to normal and diabetic controls. Glibenclamide also increased the percentages of pups with AGA and LGA, compared to the NC and DC groups, and reduced the percentage of SGA, compared to the diabetic controls. Intriguingly, the pregnant diabetic rats treated with the plant extract dose of 200mg/kg showed a maximum percentage of LGA significantly increasing the glibenclamide's percentage by 106.36% (p<0.01).

The "Fig 9" shows that maternal diabetes induced caudal regression and hematomas in fetuses. The fetuses from *A. oligophyllus* aqueous extract-treated diabetic rats displayed a more developed tail and reduced hematomas compared to those from diabetic control rats.

**Effect of the plant extract on placental mass and efficiency.** The "Fig 10A" shows that placental mass did not change significantly between the pregnant non-diabetic and diabetic rat groups. The "Fig 10B" shows that the placental efficiency significantly decreased by 40.65% (p<0.01) in pregnant diabetic control rats compared to pregnant normal control. The *A. oligophyllus* aqueous extract doses of 50 and 200mg/kg significantly increased (p<0.05) the placental efficiency respectively by 55.12% and 54.96%, while glibenclamide and the plant extract dose of 100mg/kg induced a non-significant increase (p>0.05) of 40.23% and 22.97% respectively, all compared to pregnant diabetic control "Fig 10B".

**Effect of the plant extract on histomorphological changes in placenta.** The "Fig 11" shows microphotographs of rat placental sections. Placental sections from normal rats showed glycogenic cells with glycogen deposits (Gly), trophoblastic cells (TC), giant trophoblastic cells (gTC) and a spongiotrophoblast (ST) in the basal zone. Compared to normal rats, diabetic rats showed no glycogenic cells but a very marked presence of hematomas (Hem). The *A. oligophyllus* leaves aqueous extract and glibenclamide corrected these alterations by reducing the surface area of trophoblastic hematomas. In addition, the plant extract and glibenclamide induced glycogenic cells regeneration with glycogen deposits.

## Discussion

Pregestational diabetes mellitus (PGDM) is a major risk factor for complications of pregnancy [10]. In this study, a rat model of PGDM was established to determine the effect of the aqueous extract of leaves of *Angylocalyx oligophyllus* on the hyperglycemia-induced maternal reproductive alterations as well as fetal structural and metabolic abnormalities.

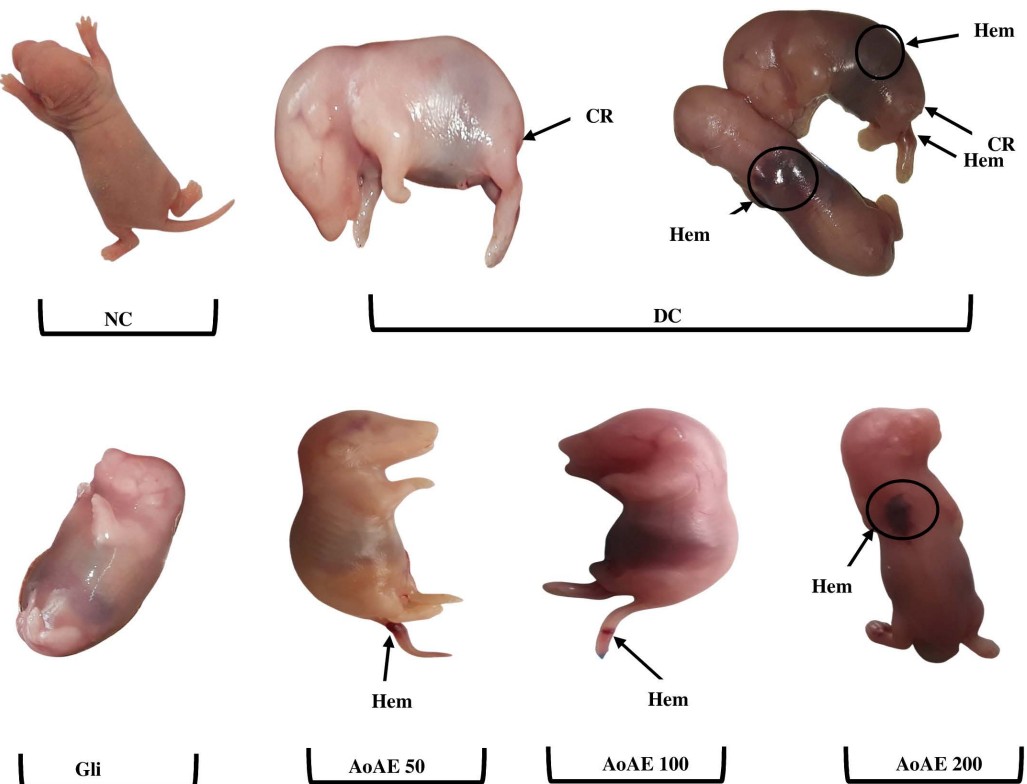

**Fig 9. External morphological malformations in fetuses from pregnant normal and untreated and treated diabetic rats.** NC = Normal control; DC = diabetic control; Gli = Pregnant diabetic rats treated with glibenclamide; *AoAE50, AoAE100* and *AoAE200* = Pregnant diabetic rats treated with *Angylocalyx oligophyllus* aqueous extract at the doses of 50, 100 and 200 mg/kg, respectively; CR = Caudal Regression; Hem = Haematomas.

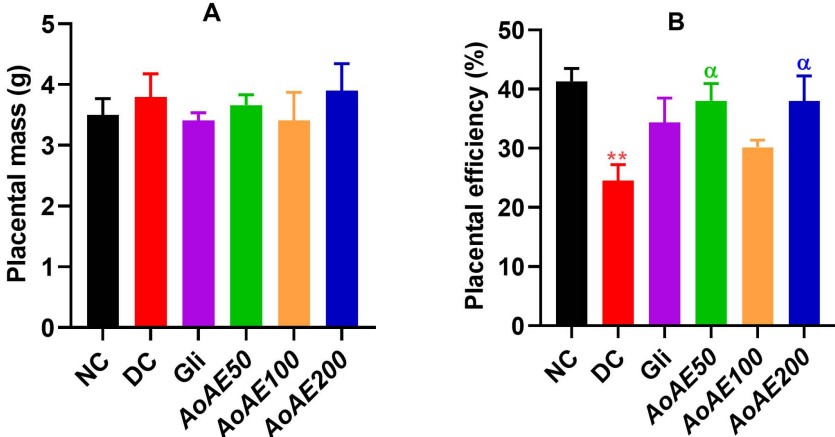

**Fig 10. Placental mass (A) and efficiency (B) in pregnant non-diabetic and, pregnant untreated and *A. oligophyllus* extract-treated diabetic rats.** Values are expressed as mean ± SEM; n = 5–8 per group; $^*p < 0.05$, = significant difference compared to NC; $^{\alpha}p < 0.05$, $^{\alpha\alpha}p < 0.01$ = significant difference compared to DC; $^{\beta\beta}p < 0.01$ = significant difference compared to Glib; NC = Normal control; DC = diabetic control; Gli = Pregnant diabetic rats treated with glibenclamide; *AoAE50, AoAE100* and *AoAE200* = Pregnant diabetic rats treated with *Angylocalyx oligophyllus* aqueous extract at the doses of 50, 100 and 200 mg/kg, respectively.

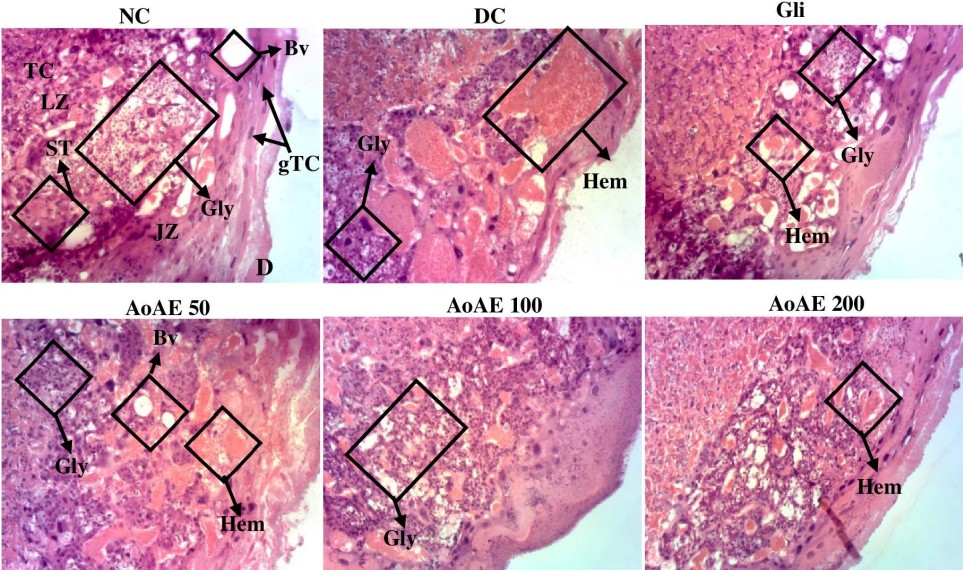

**Fig 11. Microphotography of placenta sections (haematoxylin-eosin x100) from pregnant normal and untreated and treated diabetic rats.**
NC = normal control; DC = diabetic control; Gli = Pregnant diabetic rats treated with glibenclamide; *AoAE50, AoAE100* and *AoAE200* = Pregnant diabetic rats treated with *Angylocalyx oligophyllus* aqueous extract at the doses of 50, 100 and 200 mg/kg, respectively; Gly = Glycogen, Hem = Haematoma, Bv = Blood vessels, Agly = Absence of glycogen, ST = Spongiotrophoblast, TC = Trophoblastic cells, gTC = giant Trophoblastic cells; LZ = Labyrinth Zone; JZ = Junctional Zone; D = Decidua.

The phytochemical investigation of the aqueous extract of the leaves of *A. oligophyllus* revealed the presence of polyphenols/flavonoids, terpenoids and alkaloids. The quantitative evaluation indicated that flavonoids represent nearly 70% of polyphenols. A large body of literature links these chemical classes to the diabetes mellitus treatment and even the improvement of maternal and fetal outcomes in maternal diabetes [75–78]. Coming from the same harvesting place and the same host tree, the plant material used in this study most likely has the same phytoconstituents as those reported by Wakeu Kweka et al. [38,39]. These are the flavonoid/isoflavone formononetin, terpenoids ursolic acid, betulinic acid, lupeol and lupenone, and the phytosterol β-sitosterol. According to the literature, these compounds display promising anti-diabetic properties [42,43,45,79–81] and thus, could be responsible, at least in part, for the improvement of the blood glucose control and mater-fetal outcomes observed.

Maternal hyperglycemia (diabetes, impaired fasting glucose, and impaired glucose tolerance) represents a significant health risk to the mother and the fetus [82]. It is a reflect of the severity of insulin secretory defects and/or insulin resistance, but also a poor control of maternal glucose levels. Thus, to improve pregnancy health and reduce the risk of adverse outcomes in a pre-existing T1DM, a tight control of maternal fasting and postprandial blood glucose is recommended [14][83]. To achieve optimal glucose levels, one of the well-known options is to prevent diet-dependent blood glucose rise (postprandial hyperglycemia) by inhibiting α-amylase activity [84]. The aqueous extract of the leaves of *A. oligophyllus* displayed a marked concentration-dependent α-amylase inhibitory activity with an $IC_{50}$ value closer to that of acarbose, a well-known post-prandial acting antidiabetic drug. This result suggests the capacity of the extract to delay the digestion and absorption of sugars by intestine by inhibiting digestive enzymes. The phytoconstituents identified in *A. oligophyllus* [38,39] and including formononetin, ursolic acid, β-sitosterol, betulinic acid, lupeol and lupenone have been reported to induce a marked α-amylase inhibitory activity [85–89]. Insulin is crucial for glucose uptake in tissues. Accordingly, the loss of insulin-producing pancreatic β-cells results in elevated blood glucose levels and impairment of glucose tolerance [82]. In other words, abnormal glucose tolerance is the result of a decreased insulin secretion owing to reduced

islet cells' number. Therefore, the reduction in pancreatic islets' size and islet cells' number noted in pregnant diabetic rats could explain the impaired glucose tolerance also observed in these animals. In that context of T1DM, the aqueous extract of *A. oligophyllus*, by increasing the pancreatic islets' size and islet cells' number, likely enhances insulin secretion and thus, improves glucose tolerance. The physiopathology of insulin resistance (IR) in T1DM is complex, involving glucose and lipid toxicity, low-to-moderate-grade inflammation, mitochondrial dysfunction, and oxidative stress [90,91]. A gradual IR also develops in pregnancy to ensure an adequate supply of fetus in nutrients and oxygen to its rapid growth [92]. The extract of *A. oligophyllus* improved insulin sensitivity indicating its capacity to mitigate oxidative stress associated with glucose and lipid toxicity. Insulin deficiency and insulin resistance contribute to a chronic hyperglycemia-induced oxidative stress commonly involved in pathophysiological mechanisms of feto-maternal complications. Overproduction of ROS, a common feature of a chronic hyperglycemia environment, is the consequence of a deficiency and/or resistance to insulin in PGDM. The extract of *A. oligophyllus* displayed marked antioxidant (FRAP assay) and radical scavenging (DPPH, ABTS$^+$, HO) activities displayed *in vitro*, suggesting its capacity to scavenge ROS and strength the body antioxidant defense systems. This could improve feto-maternal outcomes, including maternal reproductive health. While DPPH, ABTS and FRAP assays are appropriate for initial screening since they measure antioxidant capacity without specific cellular compartment targeting, the use of complementary methods such as DCFH-DA and MitoSOX that respectively and specifically measure cytosolic ROS and mitochondrial superoxide, would have been very helpful to know in which cellular compartment the extract acts. Globally, to modulate the blood glucose levels, the bioactive compounds present in the extract of *A. oligophyllus* including formononetin, ursolic acid, betulinic acid, lupeol, lupenone and β-sitosterol inhibit the digestive enzyme (α-amylase) activity, and by mitigating oxidative stress-induced tissue cell damages and metabolic dysregulation to enhance insulin secretion and improve insulin sensitivity.

Maternal body weight is an important physiological parameter that can provide information on mother health and fetal growth [93]. In other words, appropriate weight gain during pregnancy is crucial for the health of both the mother and the developing fetus. Gestational weight gain is a reflect of multiple characteristics, including maternal fat accumulation, fluid expansion, and the growth of the fetus, placenta, and uterus [94]. These parameters or at least some can be modified in the event of coexisting pathologies during pregnancy such as pregestational diabetes, thus affecting the proper development of pregnancy [95]. Our results showed a gain in gestational weight in both non-diabetic and diabetic pregnant rats throughout pregnancy, increment being lesser in type 1 diabetic animals. The reduced body weight observed in diabetic control animals was associated with low abdominal and peri-ovarian fat mass, small fetal mass for gestational age (SGA), and a reduced number of born fetuses, low uterine mass, while the placental mass was not different between the groups. These variations are in line with previous reports [96–98]. In STZ-induced T1DM rodent models, insulin deficiency and the uncontrolled hyperglycemia reach catabolic state characterized by low leptin levels, lipolysis and secretion of free fatty acids that move to the liver for triglycerides and LDL-cholesterol production [99–104]. In T1DM, patients with poor glycemic control usually displayed increased triglycerides and LDL-cholesterol [105]. In the present study, impaired lipid profile (increased triglycerides, total and LDL-cholesterol, reduced HDL-cholesterol) has been recorded in pregnant diabetic control rats. The administration of the extract of *A. oligophyllus* enhanced abdominal and peri-ovarian fat masses, and body weight probably by blocking lipase in adipose tissue and thus, favoring lipid storage (an antilipolytic effect). Moreover, the extract improved dyslipidemia (reduced total and LDL-cholesterol, reduced triglycerides, increased HDL-cholesterol), suggesting its capacity to reduce lipid efflux mechanisms and improve insulin signaling. Moreover, the extract more likely promotes LDL catabolism by increasing LDL-receptor expression and activity, reduces chylomicron production and enhances their catabolism, and inhibits triglyceride synthesis in liver and other tissues. HDL has anti-inflammatory, anti-oxidant and anti-apoptotic effects [105]. By removing lipid peroxides from oxidized LDL and cell membranes, HDL-cholesterol displays anti-oxidative properties. Endothelial lipase reduces its concentration and changes its properties [106]. The compounds present in the extract in addition to show a direct anti-oxidant and anti-inflammatory effect, could block endothelial lipase, rise HDL-cholesterol levels that thereafter will contribute to the global anti-inflammatory and anti-oxidant effects, crucial in managing T1DM and mitigating complications.

The structure and efficiency of the placenta play a key role in the quality of fetal-maternal exchanges and thus condition the survival, growth and morphology of the fetus [107,108]. Fetal growth restriction is frequent in type 1 PGDM [109]. By damaging placentation (structure and function), maternal hyperglycemia-induced oxidative stress disrupts the transfer of essential nutrients and oxygen leading to a fetal growth restriction, preterm birth and infant mortality [28][110,111]. Uteroplacental malperfusion or insufficiency, a consequence of a decreased or abnormal uterine artery blood flow, has been frequently associated with stillbirths and reduced fetal weight [112–114]. In the present study, the decrease in placental efficiency in pregnant diabetic control rats was associated with a high incidence of small weight for gestational age (SGA), fetal resorption, stillbirths and congenital malformations (caudal/tail regression and hematomas). The treatment of these animals with the aqueous extract of *A. oligophyllus* at all the tested doses led to the predominance of live fetuses and those with adequate weight for gestational age (AGA). A more developed tail and reduced hematomas were also observed. Through placental vasculopathic abnormalities and microvascular damages in the fetus, type 1 PGDM increases the risk of localized bleeding and hematoma formation. Placental structure and efficiency, and embryogenesis from the pre-implantation phase to fetal development are altered by the hyperglycemia-induced chronic oxidative stress and inflammation [[28],112–116]. This information suggests that the extract through its antioxidant chemicals protect placenta and fetal microvasculature against damages that lead to placental insufficiency, and fetal resorption, restriction, death, hemorrhage and hematoma. Impaired reproductive performance is a well-known consequence of T1DM in many mammalian species, including humans through immune and metabolic disorders [117]. In the present study, pregnant STZ-induced diabetic control rats showed decreased uterine horn mass, pregnancy percentage, and number of corpora lutea, implantation sites, and live pups' number, along with increased post-implantation loss, fetal resorptions, and number of dead fetuses compared to pregnant normal rats. Insulin plays an important role in maintaining the normal function of the hypothalamic-pituitary-gonadal axis [118]. Its deficiency, hyperglycemia and low leptin (due to lipoatrophy) that characterized T1DM, inhibit the expression of kisspeptin in hypothalamic neurons and blunt gonadotropin-releasing hormone (GnRH) release that lowers gonadotropin levels [119,120]. The low levels of gonadotropins, especially FSH, also impaired glucose tolerance due to insufficient insulin secretion [121]. The disruption of gonadotropin (FSH and LH) production promotes ovarian multi-follicular atresia usually associated with lower estradiol production (due to the reduction of granulosa cells through apoptosis and autophagy) and decrease in corpora lutea count [122–124]. The aqueous extract of *A. oligophyllus* increased uterine horn mass and gestation percentage, and prevented the post-implantation loss, fetal death and resorption. These results indicate the capacity of the extract to alleviate the metabolic disorder, placental malperfusion, maternal immunologic intolerance towards fetus and the hypothalamic-pituitary-ovarian axis alteration induced by chronic oxidative stress and inflammation under by hyperglycemia in T1DM.

It therefore counteracts the metabolic disorder, placental malperfusion, maternal immunologic intolerance towards fetus and the hypothalamic-pituitary-ovarian axis alteration induced by a chronic oxidative stress and inflammation

Pregnancy is the most intense physiological alteration in energy metabolism that women experience in their lifetime [125]. Profound changes therefore occur in multiple organs including liver. It is a crucial metabolic organ whose dysfunction occurs in 3% of pregnancies [126]. T1DM can induce profound hepatocyte ultrastructural alterations and cell apoptosis through oxidative stress and an aberrant inflammatory response [127–132]. Accordingly, maternal pre-existing diabetes obviously increases the risk of pregnancy-related liver injury. Aspartate aminotransferase (AST) and alanine aminotransferase (ALT) are important biomarkers frequently used to evaluate hepatocyte's damage. However, studies indicated a more predictive value in liver diseases of AST/ALT ratio [133,134]. A high AST/ALT ratio indicates advanced liver damage [135]. In our results, elevated serum values of AST, ALT and AST/ALT were observed in pregnant diabetic rats as compared to the non-pregnant controls. All these variations suggest liver damage. The microphotographs of liver sections of untreated pregnant diabetic rats showed an extensive immune/inflammatory cell infiltration and severe vascular congestion not present in pregnant normal rats. These results suggest that the mentioned alterations are mainly due to diabetes. Non-infectious liver stress and injury, regardless the cause, manifests as sterile inflammatory response marked

by immune/inflammatory cell infiltration [136,137]. This ubiquitous response occurs to a high degree in the liver and also results in high levels of tissue damage after development of metabolic syndrome [138]. In their study, Barssotti et al. [139] linked the vessel congestion in the liver of STZ-induced diabetic mice to the presence of fibrotic tissue impairing blood flow. The high levels of serum AST, ALT and AST/ALT as well as the liver inflammation and vascular congestion were markedly decreased in pregnant diabetic rats after *A. oligophyllus* treatment, indicating that it can alleviate liver injury of STZ-induced type 1 diabetic rats. On the other hand, in the present study diabetic pregnant rats exhibited a hepatomegaly. This is likely due to either a metabolic dysfunction-associated steatosis liver disease or a glycogenic hepatopathy. These two diseases are known to be the most common cause of hepatomegaly and elevated liver enzymes in the general population and in T1DM [140]. Compared to the non-diabetic pregnant controls, a liver accumulation of malondialdehyde (MDA), an oxidative damage product, was observed in pregnant diabetic animals. Contrary to earlier studies that reported reduced antioxidants (SOD, CAT and GSH) levels in diabetic pregnant rats [141,142], our results showed significant increases in antioxidant biomarkers' levels suggesting an oxidative stress state where body is attempting to mitigate oxidative damage. Such variations are usually observed at the early stages of diabetes and decline as it progresses [143]. During this period, antioxidants rise to attempt to counteract oxidative damage. The treatment of pregnant diabetic rats with the aqueous extract of *A. oligophyllus* decreased liver MDA content and antioxidant levels to values close to that observed in non-diabetic pregnant rats. The result suggests that in addition to reduce oxidative stress and lipid peroxidation, the extract also inhibit excessive production of antioxidants and help restore a balance in antioxidant system.

The STZ-induced diabetes model (35 mg/kg) leads to a partial destruction of pancreatic beta cells and can serve as a model to evaluate the pancreatic effect of a substance that controls blood sugar on insulin secretion, cellular damage repair, and cellular regeneration. Thus, a standard antidiabetic agent that stimulates insulin secretion and regenerates beta cells, such as glibenclamide, would be a good comparator for determining, at least in part, the mechanism of action of the substance being studied. Although the present study lacks data on insulin levels, the results showed that the *A. oligophyllus* extract not only controlled hyperglycemia and hyperlipidemia in pregnant diabetic rats but also increased the mass of their pancreatic cells. This indicates cellular protection and/or regeneration, particularly of the beta cells, likely promoted by the plant's observed antioxidant effects. These effects of the plant were found to be superior or comparable to those of glibenclamide. The ease and route of administration, reduced cost, and acceptance of oral antidiabetic agents like glibenclamide have encouraged a considerable increase in its use for managing diabetes mellitus and, in some contexts, it is the first-line treatment option for gestational diabetes requiring medication-based management [144]. Although glibenclamide has long been a considerable alternative to insulin, recent studies have reported that it can cross the placenta, stimulate fetal insulin secretion, and increase the risks of macrosomia (LGA) and neonatal hypoglycemia in a dose-dependent manner [144,145]. However, it has been reported that glibenclamide doses of 5 and 20 mg/kg do not cause any fetal alterations [146], which suggests that the 10 mg/kg glibenclamide dose used in the present study would have very few deleterious risks, at least within the time limit of the study's use, despite increasing fetal weight. Interestingly, these data on the dose-dependent effects of glibenclamide prompt a relevant consideration regarding the dose-response effect of the *A. oligophyllus* extract on the percentage of LGA and the need for a further investigation into the plant's fetal and postnatal effects. Indeed, the high percentage of LGA recorded at the 200 mg/kg extract dose compared to glibenclamide (46.43% versus 22.5%) reinforces the hypothesis of an increased risk of overweight at birth linked to high doses of this plant extract, as previously reported by Tenezogang et al. [49] in a reprotoxicity study of *A. oligophyllus*. This suggests that low doses of the plant extract would be appropriate for managing diabetes during pregnancy.

Although the model used in this study is well-known and reliable, and *A. oligophyllus* is used empirically for pregnancy improvement in case of diabetes, further studies are needed for translating findings. On the other hand, the most critical limitation of this study is the absence of a non-pregnant diabetic group to distinguish pregnancy-specific versus general metabolic effects of the extract.



## Conclusion

The present study revealed that diabetes negatively impacts fertilization, nidation and the course of pregnancy. However, aqueous extract from *A. oligophyllus* leaves, administered from day 1 of gestation, maintained pregnancy, improved feto-maternal outcomes and reproduction in pregnant diabetic rats through the antioxidant, antidiabetic and anti-inflammatory effects of its bioactive molecules such as formononetin, ursolic acid, betulinic acid, lupeol, lupenone and β-sitosterol.

## Supporting information

**S1 Data.  Minimal Data Set From Pone-D-25-21592.**
(PDF)

## Acknowledgments

We would like to express our sincere thanks to the Alexander Von Humboldt Foundation for awarding the equipment grant to one of the authors which enabled part of this work to be carried out.

## Author contributions

**Conceptualization:** Christian Tenezogang Takoukam, Marie Claire Tchamadeu, Dieudonné Massoma Lembè.

**Data curation:** Christian Tenezogang Takoukam, Marie Claire Tchamadeu.

**Formal analysis:** Christian Tenezogang Takoukam, Sylvin Benjamin Ateba, William Yousseu Nana, Calvin Bogning Zangue, Pascal Emmanuel Owona.

**Investigation:** Christian Tenezogang Takoukam, Quelie Selakong Nzekuie, Armel-Kevin Pechi Fotso, Ahmadou Hassimatou, Calvin Bogning Zangue, Pascal Emmanuel Owona.

**Methodology:** Christian Tenezogang Takoukam, Marie Claire Tchamadeu, Dieudonné Massoma Lembè.

**Supervision:** Dieudonné Massoma Lembè.

**Validation:** Marie Claire Tchamadeu, Dieudonné Massoma Lembè.

**Visualization:** Christian Tenezogang Takoukam, Marie Claire Tchamadeu, Dieudonné Massoma Lembè.

**Writing – original draft:** Christian Tenezogang Takoukam.

**Writing – review & editing:** Marie Claire Tchamadeu, Sylvin Benjamin Ateba, William Yousseu Nana, Modeste Wankeu-Nya, Alain Bertrand Dongmo, Dieudonné Massoma Lembè.

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
