## [Decision Letter · Decision Letter 0]

26 Jun 2025

Dear Dr. TCHAMADEU,

Thank you for submitting your manuscript to PLOS ONE. We appreciate the opportunity to evaluate your work and commend the effort invested in this research. After careful assessment, our editorial team and peer reviewers’ constructive feedback that your study requires revisions to meet the journal’s standards. Below, please find the reviewers’ comments for your reference.

We look forward to receiving your revised manuscript.

Kind regards,

Muhammad Zeeshan Bhatti, Ph.D

Academic Editor

PLOS ONE

Journal Requirements:

2. We note that your Data Availability Statement is currently as follows: All relevant data are within the manuscript and in Supporting Information files.

Additional Editor Comments:

The plant species name, *Angylocalyx oligophyllus* , must be italicized consistently throughout the manuscript.In the introduction section “The literature review should be expanded to provide stronger context for the study”.The description of methods is unclear “The study utilised the in-vitro and in-vivo experimental methods conducted over 2-months, with the in-vivo study conducted over 42 days”. Clarify the overlap (if any) between these timelines.The referenced ethical approval (2021) appears unrelated to this study. Provide the specific approval certificate (including protocol number, date, and approving institution) for the current work.The Methods section lacks critical information: Geographic location and habitat where *A. oligophyllus* leaves were collected, and botanical identification (voucher specimen number, herbarium depository, and authority who confirmed the species).Discussion section: Strengthen the discussion by comparing your findings with similar studies on plant-derived anti-diabetic/antioxidant agents. Cite recent literature to contextualize your resultsAuthor contribution: Revise this section to be concise and specific. Avoid overly verbose descriptions.The concentrations used in Figure 2C differ from those in Figures 2A/B. Justify this discrepancy or standardize concentrations across all panels for coherence.Ensure all data underlying the findings are fully available per *PLOS ONE* ’s policy.Consider professional language editing to refine grammar and syntax where needed.

We invite you to submit a revised version addressing these concerns, accompanied by a detailed point-by-point response to each comment. While we acknowledge that revisions may require additional work, we believe these suggestions will enhance the clarity, rigor, and impact of your paper.

Thank you again for considering *PLOS ONE*  for your work. We look forward to your response.

Reviewers' comments:

Reviewer's Responses to Questions

**Comments to the Author**

1. Is the manuscript technically sound, and do the data support the conclusions?

Reviewer #1: Yes

Reviewer #2: Yes

Reviewer #3: Yes

2. Has the statistical analysis been performed appropriately and rigorously?

Reviewer #1: Yes

Reviewer #2: I Don't Know

Reviewer #3: Yes

3. Have the authors made all data underlying the findings in their manuscript fully available?

Reviewer #1: Yes

Reviewer #2: Yes

Reviewer #3: Yes

4. Is the manuscript presented in an intelligible fashion and written in standard English?

Reviewer #1: Yes

Reviewer #2: Yes

Reviewer #3: Yes

**Reviewer #1:**  The manuscript provides important insights into the therapeutic potential of Angylocalyx oligophyllus in ameliorating diabetic complications during pregnancy. The study is methodologically sound, employing both in vitro and in vivo approaches, and addresses a timely and clinically relevant topic given the global rise in pregestational diabetes and the demand for safer, plant-based therapeutic options.

The authors effectively demonstrate that the plant extract exhibits notable antioxidant and anti-α-amylase activity, which correlates with improved maternal metabolic parameters and fetal development. Key findings—including reduced blood glucose levels, improved lipid profiles, preservation of pancreatic β-cell mass, and normalization of fetal outcomes—support the therapeutic promise of the extract. The thorough evaluation of reproductive and fetal parameters adds further strength to the study.

That said, I recommend a few revisions to enhance the scientific rigor and clarity of the manuscript:

Bioactive Compounds and Mechanism:

While the authors refer to general classes of phytochemicals (e.g., polyphenols, flavonoids, tannins), it would be beneficial to provide specific examples—such as quercetin, catechin, or kaempferol—and discuss their known bioactivities in the context of diabetes and oxidative stress. Referencing prior phytochemical analyses of Angylocalyx oligophyllus or conducting additional chemical profiling would substantiate the mechanistic rationale and improve the translational relevance of the findings.

Oxidative Stress and In Vitro Assessment:

Hyperglycemia leads to excessive intracellular glucose uptake, which increases the flux through glycolysis and the TCA cycle, thereby generating large amounts of NADH and FADH₂. This oversupply saturates the mitochondrial electron transport chain and results in the leakage of electrons that reduce oxygen to form superoxide radicals (O₂⁻), making this the central mechanism of diabetes-associated oxidative stress. In this context, the manuscript’s use of DPPH and ABTS assays to evaluate antioxidant activity is appropriate for initial screening. However, to better validate the antioxidant efficacy of the extract under more biologically relevant conditions, additional assays—such as DCFH-DA for intracellular ROS or MitoSOX™ Red for mitochondrial superoxide detection—should be considered or at least discussed. Incorporating such cell-based antioxidant assessments, or elaborating on this methodological limitation in the Discussion, would significantly improve the depth and translational potential of the antioxidant claims.

Mechanistic Pathways in Glucose Regulation:

A more detailed discussion is warranted on how the extract modulates blood glucose levels. Does it primarily act through inhibition of digestive enzymes, enhancement of insulin secretion, improvement of insulin sensitivity, or by counteracting oxidative stress? Given the known role of oxidative stress in diabetic complications—particularly during pregnancy—clarifying the primary pathways involved and identifying the key bioactive compounds will strengthen the mechanistic framework and help guide future investigations.

Organ Protection and Toxicity:

The manuscript mentions reductions in markers of organ damage (e.g., transaminases), but lacks sufficient discussion of hepatic or renal protective effects. Including additional data or referencing relevant histopathological or biochemical analyses could clarify whether the extract offers protective effects to maternal organs beyond metabolic regulation.

In summary, this manuscript offers promising preclinical evidence for the use of Angylocalyx oligophyllus in the management of gestational diabetes. Addressing the points above will improve the depth and clarity of the study, thereby enhancing its scientific contribution and clinical relevance. Upon revision, this work is suitable for publication and will be of interest to researchers in the fields of phytotherapy, reproductive toxicology, and diabetes research.

**Reviewer #2:**  This study investigated the effects of Angylocalyx oligophyllus leaf aqueous extract on diabetes-induced metabolic, reproductive, and fetal developmental disorders in pregnant diabetic rats. The extract showed significant in vitro anti-α-amylase and antioxidant activities. Diabetic pregnant rats treated with the extract from gestation days 1 to 19 showed reduced blood glucose, cholesterol, triglycerides, and liver enzymes, while HDL-cholesterol, estradiol levels, pancreatic cells, and implantation sites increased. The extract also reduced post-implantation losses, fetal malformations, and improved fetal weight and survival. These findings suggest that A. oligophyllus supplementation may help prevent diabetes-related reproductive complications during pregnancy. The manuscrit is well writen. However, in the introduction section, it should be given information about antioxidants.

**Reviewer #3: ** This study explores the therapeutic potential of Angylocalyx oligophyllus aqueous leaf extract in mitigating the adverse effects of pregestational diabetes in pregnant rats. Diabetic rats exhibited significant metabolic disturbances, reproductive failures, and fetal abnormalities, including hyperglycemia, dyslipidemia, reduced estradiol, increased fetal loss, and malformations. Treatment with the plant extract improved these conditions by restoring glycemic and lipid balance, enhancing reproductive outcomes (such as implantation and live births), increasing estradiol levels and pancreatic cell counts, and reducing fetal deformities. The extract demonstrated notable antioxidant and α-amylase inhibition properties, suggesting that it could be a promising natural agent for preventing diabetes-related reproductive complications during pregnancy. Specific comments:

1. The manuscript implies a preventive role of Angylocalyx oligophyllus (AO) during diabetic pregnancy, but the primary hypothesis and objectives need clearer articulation at the end of the Introduction.

2. Phytochemical Quantification While qualitative phytochemistry was conducted, consider quantifying key constituents (e.g., polyphenols, flavonoids, saponins) using standard calibration curves to better relate bioactivity to composition.

3. In Vitro to In Vivo Link The translational bridge between the in vitro antioxidant/α-amylase activity and in vivo outcomes is weak. Could the authors discuss how these mechanisms might directly impact reproductive health?

4. Statistical Interpretation The statement “significant (p<0.5 – p<0.001)” is ambiguous. Please confirm the actual p-values used in each comparison and ensure consistent reporting across the manuscript.

5. Dose Justification The empirical basis for the 50, 100, and 200 mg/kg doses is mentioned briefly. Consider referencing more explicit toxicity or dose-response studies validating this range.

6. Histopathological Evaluation Were histological sections of the placenta, pancreas, or fetal tissues conducted? Their inclusion could strengthen the mechanistic implications of AO’s protective effect.

7. Control Group Clarification Was a non-pregnant diabetic group included for comparison? If not, how do we distinguish pregnancy-specific versus general metabolic effects of the extract?

8. Standard Drug Control Glibenclamide was used as a comparator, but its reproductive toxicity is debated. Could the authors justify this choice and discuss limitations?

9. Terminology and Style Phrases like "borned-living fetuses" should be revised for clarity. A thorough language and grammar revision is needed throughout, especially in Results and Discussion.

10. Ethnobotanical Context While the local use of AO is cited, the ethnobotanical rationale behind using it for pregnancy-related complications could be better developed—possibly supported by interviews or broader literature.

**Do you want your identity to be public for this peer review?** For information about this choice, including consent withdrawal, please see our Privacy Policy

Reviewer #1: No

Reviewer #2: No

Reviewer #3: **Yes: ** Yung-Hsiang Chen

---

## [Author Response · Author response to Decision Letter 1]

16 Sep 2025

RESPONSE TO THE EDITOR AND REVIEWERS

Journal Requirements:

R: Changes have been made accordingly

2. We note that your Data Availability Statement is currently as follows: All relevant data are within the manuscript and in Supporting Information files.

R: Values behind the means and those used to build graphs have been provided in an excel file as Supporting Information (SI3) files and added as recommended

Additional Editor Comments:

1. The plant species name, Angylocalyx oligophyllus, must be italicized consistently throughout the manuscript.

R: The plant name has been checked and italicized throughout the manuscript

2. In the introduction section “The literature review should be expanded to provide stronger context for the study”.

R: Changes have been made accordingly

3. The description of methods is unclear “The study utilised the in-vitro and in-vivo experimental methods conducted over 2-months, with the in-vivo study conducted over 42 days”. Clarify the overlap (if any) between these timelines.

R: The mentioned sentence has been reformulated

4. The referenced ethical approval (2021) appears unrelated to this study. Provide the specific approval certificate (including protocol number, date, and approving institution) for the current work.

R: This work is part of TENEZOGANG Ph.D project started in 2021 after obtaining in April 2021 an ethical approval from the Institutional Ethics Committee of the University of Douala as indicated in the manuscript

5. The Methods section lacks critical information: Geographic location and habitat where A. oligophyllus leaves were collected, and botanical identification (voucher specimen number, herbarium depository, and authority who confirmed the species).

R: Changes have been made

6. Discussion section: Strengthen the discussion by comparing your findings with similar studies on plant-derived anti-diabetic/antioxidant agents. Cite recent literature to contextualize your results

R: Discussion section has been completely rewritten to adress this comment

7. Author contribution: Revise this section to be concise and specific. Avoid overly verbose descriptions.

R: changes have been made

8. The concentrations used in Figure 2C differ from those in Figures 2A/B. Justify this discrepancy or standardize concentrations across all panels for coherence.

R: Concentrations have been standardized across all panels for coherence

9. Ensure all data underlying the findings are fully available per PLOS ONE’s policy.

R: All data underlying the findings are fully available

10. Consider professional language editing to refine grammar and syntax where needed.

R: We rewrote many parts of the manuscript to try to refine grammar and syntax as recommended

5. Review Comments to the Author

Reviewer #1: The manuscript provides important insights into the therapeutic potential of Angylocalyx oligophyllus in ameliorating diabetic complications during pregnancy. The study is methodologically sound, employing both in vitro and in vivo approaches, and addresses a timely and clinically relevant topic given the global rise in pregestational diabetes and the demand for safer, plant-based therapeutic options.

The authors effectively demonstrate that the plant extract exhibits notable antioxidant and anti-α-amylase activity, which correlates with improved maternal metabolic parameters and fetal development. Key findings—including reduced blood glucose levels, improved lipid profiles, preservation of pancreatic β-cell mass, and normalization of fetal outcomes—support the therapeutic promise of the extract. The thorough evaluation of reproductive and fetal parameters adds further strength to the study.

That said, I recommend a few revisions to enhance the scientific rigor and clarity of the manuscript:

Bioactive Compounds and Mechanism:

While the authors refer to general classes of phytochemicals (e.g., polyphenols, flavonoids, tannins), it would be beneficial to provide specific examples—such as quercetin, catechin, or kaempferol—and discuss their known bioactivities in the context of diabetes and oxidative stress. Referencing prior phytochemical analyses of Angylocalyx oligophyllus or conducting additional chemical profiling would substantiate the mechanistic rationale and improve the translational relevance of the findings.

R: Changes have been made accordingly

Oxidative Stress and In Vitro Assessment:

Hyperglycemia leads to excessive intracellular glucose uptake, which increases the flux through glycolysis and the TCA cycle, thereby generating large amounts of NADH and FADH₂. This oversupply saturates the mitochondrial electron transport chain and results in the leakage of electrons that reduce oxygen to form superoxide radicals (O₂⁻), making this the central mechanism of diabetes-associated oxidative stress. In this context, the manuscript’s use of DPPH and ABTS assays to evaluate antioxidant activity is appropriate for initial screening. However, to better validate the antioxidant efficacy of the extract under more biologically relevant conditions, additional assays—such as DCFH-DA for intracellular ROS or MitoSOX™ Red for mitochondrial superoxide detection—should be considered or at least discussed. Incorporating such cell-based antioxidant assessments, or elaborating on this methodological limitation in the Discussion, would significantly improve the depth and translational potential of the antioxidant claims.

R: We did not perform DCFH-DA and MitoSOX assays. To address your comment, this methodological limitation has been incorporated in the discussion

Mechanistic Pathways in Glucose Regulation:

A more detailed discussion is warranted on how the extract modulates blood glucose levels. Does it primarily act through inhibition of digestive enzymes, enhancement of insulin secretion, improvement of insulin sensitivity, or by counteracting oxidative stress? Given the known role of oxidative stress in diabetic complications—particularly during pregnancy—clarifying the primary pathways involved and identifying the key bioactive compounds will strengthen the mechanistic framework and help guide future investigations.

R: A more detailed discussion related to this comment has been written (the third paragraph of the discussion)

Organ Protection and Toxicity:

The manuscript mentions reductions in markers of organ damage (e.g., transaminases), but lacks sufficient discussion of hepatic or renal protective effects. Including additional data or referencing relevant histopathological or biochemical analyses could clarify whether the extract offers protective effects to maternal organs beyond metabolic regulation.

R: To avoid over-speculation, the part of the discussion referring to the protective effect of the extract on the kidney has been removed because no results were presented that could support such a statement. Furthermore, regarding the liver, in addition to data on the relative weight, AST and ALT transaminase levels, histological sections and data on redox status (MDA and GSH contents, CAT and SOD activities) have been added and discussed.,

Reviewer #2: This study investigated the effects of Angylocalyx oligophyllus leaf aqueous extract on diabetes-induced metabolic, reproductive, and fetal developmental disorders in pregnant diabetic rats. The extract showed significant in vitro anti-α-amylase and antioxidant activities. Diabetic pregnant rats treated with the extract from gestation days 1 to 19 showed reduced blood glucose, cholesterol, triglycerides, and liver enzymes, while HDL-cholesterol, estradiol levels, pancreatic cells, and implantation sites increased. The extract also reduced post-implantation losses, fetal malformations, and improved fetal weight and survival. These findings suggest that A. oligophyllus supplementation may help prevent diabetes-related reproductive complications during pregnancy. The manuscrit is well writen. However, in the introduction section, it should be given information about antioxidants.

R: The introduction section has been completely rewritten to deal with the recommendation

Reviewer #3: This study explores the therapeutic potential of Angylocalyx oligophyllus aqueous leaf extract in mitigating the adverse effects of pregestational diabetes in pregnant rats. Diabetic rats exhibited significant metabolic disturbances, reproductive failures, and fetal abnormalities, including hyperglycemia, dyslipidemia, reduced estradiol, increased fetal loss, and malformations. Treatment with the plant extract improved these conditions by restoring glycemic and lipid balance, enhancing reproductive outcomes (such as implantation and live births), increasing estradiol levels and pancreatic cell counts, and reducing fetal deformities. The extract demonstrated notable antioxidant and α-amylase inhibition properties, suggesting that it could be a promising natural agent for preventing diabetes-related reproductive complications during pregnancy. Specific comments:

1. The manuscript implies a preventive role of Angylocalyx oligophyllus (AO) during diabetic pregnancy, but the primary hypothesis and objectives need clearer articulation at the end of the Introduction.

R: Changes have been made accordingly

2. Phytochemical Quantification While qualitative phytochemistry was conducted, consider quantifying key constituents (e.g., polyphenols, flavonoids, saponins) using standard calibration curves to better relate bioactivity to composition.

R: The key constituents such as polyphenols and flavonoids were quantified using calibration curves. Results are shown in table 1. Standard calibration curves have been provided in a Supporting Information file (SI3)

3. In Vitro to In Vivo Link The translational bridge between the in vitro antioxidant/α-amylase activity and in vivo outcomes is weak. Could the authors discuss how these mechanisms might directly impact reproductive health?

R: we tried to strength the link between the in vitro antioxidant/α-amylase activity and in vivo outcomes in the third paragraph of the discussion.

4. Statistical Interpretation The statement “significant (p<0.5 – p<0.001)” is ambiguous. Please confirm the actual p-values used in each comparison and ensure consistent reporting across the manuscript.

R: p-values have been correctly rewritten across the manuscript

5. Dose Justification The empirical basis for the 50, 100, and 200 mg/kg doses is mentioned briefly. Consider referencing more explicit toxicity or dose-response studies validating this range.

R: This text portion has been rewritten in order to be more explicit and more understandable in the choice of the doses

6. Histopathological Evaluation Were histological sections of the placenta, pancreas, or fetal tissues conducted? Their inclusion could strengthen the mechanistic implications of AO’s protective effect.

R: Histopathological evaluation were conducted on histological sections of pancreas and liver as maternal tissues and of placenta as materno-fetal tissue.

7. Control Group Clarification Was a non-pregnant diabetic group included for comparison? If not, how do we distinguish pregnancy-specific versus general metabolic effects of the extract?

R: This is a very interesting remark. Unfortunately, a non-pregnant diabetic group has not been included for comparison. This constitute a critical limitation of this study. We formulate and indicate it at the end of the discussion

8. Standard Drug Control Glibenclamide was used as a comparator, but its reproductive toxicity is debated. Could the authors justify this choice and discuss limitations?

R: Justification of the standard drug: The choice of glibenclamide as the standard comparator drug was based on its ease and route of administration (oral route like the plant extract), as well as its reduced cost and better acceptance (less traumatic consumption than insulin injections). Furthermore, it is more appropriate for the present induced diabetes model (as the STZ dose of 35 mg/kg leads to a partial destruction of β-cells, leaving a number of residual cells on which glibenclamide can act as an insulin secretagogue).

Discussion of the Limits: See discussion lines 941 – 969.

9. Terminology and Style Phrases like "borned-living fetuses" should be revised for clarity. A thorough language and grammar revision is needed throughout, especially in Results and Discussion.

R: Terminology and Style Phrases have been reviewed in correct tense

10. Ethnobotanical Context While the local use of AO is cited, the ethnobotanical rationale behind using it for pregnancy-related complications could be better developed—possibly supported by interviews or broader literature.

R: In Cameroon, people of Song-Bong (Center Region) traditionally use this plant to treat eye infections and diabetes. However, ethnopharmacological information from sellers of natural medicinal products report that the leaves of A. oligophyllus are used to manage pregnancy in diabetic women.

---

## [Decision Letter · Decision Letter 1]

24 Sep 2025

Antioxidant and Anti-diabetic effects of Angylocalyx oligophyllus leaves aqueous extract in pregnant diabetic rats: Feto-maternal repercussions

PONE-D-25-21592R1

Dear Dr. TCHAMADEU,

We’re pleased to inform you that your manuscript has been judged scientifically suitable for publication and will be formally accepted for publication once it meets all outstanding technical requirements.

Kind regards,

Muhammad Zeeshan Bhatti, Ph.D

Academic Editor

PLOS ONE

Additional Editor Comments (optional):

Reviewer #1:

Reviewer #2:

Reviewer #3:

Reviewers' comments:

Reviewer's Responses to Questions

**Comments to the Author**

Reviewer #1: All comments have been addressed

Reviewer #2: All comments have been addressed

Reviewer #3: All comments have been addressed

2. Is the manuscript technically sound, and do the data support the conclusions?

Reviewer #1: Yes

Reviewer #2: Yes

Reviewer #3: Yes

3. Has the statistical analysis been performed appropriately and rigorously?

Reviewer #1: Yes

Reviewer #2: I Don't Know

Reviewer #3: Yes

4. Have the authors made all data underlying the findings in their manuscript fully available?

Reviewer #1: Yes

Reviewer #2: Yes

Reviewer #3: Yes

5. Is the manuscript presented in an intelligible fashion and written in standard English?

Reviewer #1: Yes

Reviewer #2: Yes

Reviewer #3: Yes

Reviewer #1: This manuscript demonstrates a thorough and appropriate response to Reviewer’s comments, with the content effectively incorporated into the revised version.

I believe that this revised manuscript is suitable for publication in PLOS One.

Reviewer #2: Angylocalyx oligophyllus extract reduced hyperglycemia, oxidative stress, and reproductive disorders in diabetic pregnant rats, showing protective effects on both mother and fetus.

Reviewer #3: This study explores the therapeutic potential of Angylocalyx oligophyllus aqueous leaf extract in mitigating the adverse effects of pregestational diabetes in pregnant rats. Diabetic rats exhibited significant metabolic disturbances, reproductive failures, and fetal abnormalities, including hyperglycemia, dyslipidemia, reduced estradiol, increased fetal loss, and malformations. Treatment with the plant extract improved these conditions by restoring glycemic and lipid balance, enhancing reproductive outcomes (such as implantation and live births), increasing estradiol levels and pancreatic cell counts, and reducing fetal deformities. The extract demonstrated notable antioxidant and α-amylase inhibition properties, suggesting that it could be a promising natural agent for preventing diabetes-related reproductive complications during pregnancy. The revision of the manuscript is much improved; no additional comments.

**Do you want your identity to be public for this peer review?** For information about this choice, including consent withdrawal, please see our Privacy Policy

Reviewer #1: No

Reviewer #2: No

Reviewer #3: **Yes: ** Yung-Hsiang Chen

---

## [Editor Report · Acceptance letter]

PONE-D-25-21592R1

PLOS ONE

Dear Dr. TCHAMADEU,

I'm pleased to inform you that your manuscript has been deemed suitable for publication in PLOS ONE. Congratulations! Your manuscript is now being handed over to our production team.

Kind regards,

on behalf of

Dr. Muhammad Zeeshan Bhatti

Academic Editor

PLOS ONE